# The Tbx1 ortholog *org-1* is required to establish testis stem cell niche identity in *Drosophila*

Patrick Hofe[1], Ariel Harrington[1], Tynan Gardner[2], Stephen DiNardo[2] and Lauren Anllo[1,*]

## ABSTRACT

Stem cells require signals from a cellular microenvironment known as the niche that regulates identity, location and division of stem cells. Niche cell identity must be properly specified during development to form a tissue capable of functioning in the adult. Here, we show that the Tbx1 ortholog *org-1* is expressed in *Drosophila* testis niche cells in response to Slit and FGF signals. *org-1* is expressed during niche development and is required to specify niche cell identity. *org-1* mutants specified fewer niche cells, and those cells showed disruption of niche-specific markers, including loss of the niche adhesion protein Fas3 and reduced *hedgehog* expression. We found that *org-1* expression in somatic gonadal precursors is capable of inducing formation of additional niche cells. Disrupted niche identity in *org-1* mutants caused defects in niche assembly and functionality. We found that the conserved transcription factor *islet* is expressed in response to *org-1* and show that *islet* functions downstream to mediate niche identity and assembly. This work identifies a previously unreported role for *org-1* in niche establishment.

KEY WORDS: Stem cell niche, Tbx1, Org1, Niche identity, Gonadogenesis, *Drosophila*

## INTRODUCTION

Stem cells play a vital role maintaining functional tissues. Preservation of stem cells in many tissues is controlled by the niche, a local cellular microenvironment supplying signals to maintain stem cell identity, location and division (Li and Xie, 2005). A key concept in stem cell biology is the use of diffusible signals to regulate stem cell behaviors. In models like the mammalian intestinal stem cell (ISC) niche, diffusible signals Wnt, Hh and BMP regulate stem cells (Walker et al., 2008). These signals have varying origins since there are multiple cell types that make up the ISC niche (Walker et al., 2008). Because the niche can comprise various cell lineages, identification of niche cells is often challenging. In some tissues, even the identity of the stem cells is unclear (Sato et al., 2011; Kaestner, 2019; Li et al., 2023). Thus, identifying regulators that specify niche identity during development has remained a challenge in stem cell biology.

Because of its readily identifiable, well-defined niche, we leverage the *Drosophila* testis to study how niche cells acquire

[1]Department of Biology, East Carolina University, Greenville, NC 27858, USA.
[2]Perelman School of Medicine at the University of Pennsylvania, Philadelphia, PA 19104, USA.

*Author for correspondence (anllol22@ecu.edu)

A.H., 0009-0002-3803-8218; S.D., 0000-0003-4131-5511; L.A., 0000-0001-5482-5882

identity and adopt function during development (de Cuevas and Matunis, 2011; Le Bras and Van Doren, 2006; Anllo et al., 2019; Okegbe and DiNardo, 2011; Wingert and DiNardo, 2015). The *Drosophila* testis is a paradigmatic niche in which foundational concepts in stem cell biology continue to be elucidated (Herrera and Bach, 2019; Leatherman and DiNardo, 2008, 2010; Lenhart and DiNardo, 2015; Inaba et al., 2015; Tulina and Matunis, 2001; Yamashita et al., 2003; Zheng et al., 2011). Defined cellular components of the testis niche, somatic 'hub' cells, regulate germline and somatic stem cells required to maintain sperm production throughout the adult life of the fly (Lenhart and DiNardo, 2015; Roach and Lenhart, 2024). The wealth of genetic tools available to study this niche make it an ideal model for identifying regulators of niche cell fate.

Prior to niche formation, the embryonic stage 15 *Drosophila* testis is a spherical arrangement of somatic gonadal precursors (SGPs) interspersed among and encysting germ cells (Jenkins et al., 2003; Le Bras and Van Doren, 2006). SGPs are specified from a subset of mesodermal cells adjacent to muscle precursors in embryonic parasegments 10-13 earlier in embryogenesis (Boyle and DiNardo, 1995; Boyle et al., 1997; Riechmann et al., 1998). Niche, or 'hub', cells are derived from SGPs, and are initially specified through Notch signaling (Kitadate and Kobayashi, 2010; Okegbe and DiNardo, 2011; DiNardo et al., 2011). By stage 11 of embryogenesis, SGPs contacting the primordial midgut receive Delta ligands, which initiate Notch signaling and niche specification (Okegbe and DiNardo, 2011). Continued specification, or further induction of niche cell identity after the initial acquisition of niche cell fate, is required to establish the niche. As part of continued specification, SGPs at the gonad anterior receive the Notch ligand Serrate from other gonadal cells, and SGPs at the posterior receive EGFR antagonistic signals that repress niche identity (Kitadate and Kobayashi, 2010). Defects in Notch signaling result in fewer niche cells specified (Okegbe and DiNardo, 2011; DiNardo et al., 2011; Kitadate and Kobayashi, 2010; Wingert and DiNardo, 2015). Specified niche cells downregulate the conserved transcription factor Traffic jam (Tj) in response to Notch (Wingert and DiNardo, 2015). A role for Notch in niche formation has translated to other niches in *Drosophila* and in mammals (Zamfirescu et al., 2022). Outside of Notch signaling, there is little known about how testis niche cells acquire their identity.

After the initiation of specification, embryonic niche cells are compartmentalized to the gonad anterior in a process called niche assembly (Le Bras and Van Doren, 2006; Anllo et al., 2019). During assembly, specified prospective (pro) niche SGPs move peripherally onto the gonadal extracellular matrix (ECM), where they then migrate to their anterior assembly position (Anllo et al., 2019). Our previous work revealed that extrinsic Slit and FGF signals from neighboring visceral muscle (Vm) are required for this migration (Anllo and DiNardo, 2022). At the gonad anterior, niche cells first exhibit F-actin polarization to niche-niche interfaces,

which later re-polarizes to niche-stem cell interfaces as niche cells compact into a spherical arrangement with a smooth contour (Anllo and DiNardo, 2022; Warder et al., 2024). Germline stem cells contacting the assembled niche contribute to niche morphogenesis through actomyosin networks that result in tension around the compacting hub periphery (Warder et al., 2024). The final niche expresses niche-specific markers such as *fasciclin3* and *hedgehog*, and is radially surrounded by germline stem cells (GSCs) and, later, cyst stem cells (CySCs) (Le Bras and Van Doren, 2006; Sheng et al., 2009; Sinden et al., 2012; Wingert and DiNardo, 2015). The niche makes adhesive connection to neighboring stem cells and extracellular matrix to maintain its shape, location and function (Lee et al., 2008; Tanentzapf et al., 2007; Tseng et al., 2022; Amoyel et al., 2013).

After niche establishment, the niche functions throughout adult life. The location of the niche, which is crucial for signaling to GSCs and CySCs, is maintained through Integrin-mediated connections to the ECM (Anllo et al., 2019; Lee et al., 2008; Tanentzapf et al., 2007). In its established position, the niche sends maintenance signals to its surrounding stem cells, including Unpaired (Upd) and Hedgehog (Hh). Upd activates JAK/STAT signaling in GSCs, which regulates their adhesion to the niche and oriented cell divisions (Chen et al., 2018; Leatherman and DiNardo, 2008, 2010; Tulina and Matunis, 2001; Wawersik et al., 2005; Yamashita et al., 2003). Niche-secreted Hedgehog regulates CySC self-renewal (Amoyel et al., 2013). Niche signals are important for establishing the stem cell populations, and for long-term stem cell maintenance (Sheng et al., 2009; Sinden et al., 2012).

Because Notch signaling required to specify niche cells is initiated at stage 11 of embryogenesis (Okegbe and DiNardo, 2011), and niche-specific markers such as Fas3 accumulation are not observed until stage 15 (Le Bras and Van Doren, 2006; Anllo et al., 2019), there are likely other factors influencing specification of niche cells during this period. Our work here identifies the Tbx1 ortholog *org-1* as one of those factors. Tbx1 and its orthologs play a conserved role across phyla in specifying mesodermal cell types (Marcellini et al., 2003). Like Tbx1, *org-1* encodes a transcription factor with a T-box DNA-binding domain and is a regulator of mesodermal muscle fate during embryonic development (Schaub et al., 2012; Schaub and Frasch, 2013; Boukhatmi et al., 2014). Our previous work identified that the conserved transcription factor *islet* is expressed in niche cells and required for both niche assembly and cytoskeletal polarization downstream of Vm assembly cues (Anllo and DiNardo, 2022). *islet* expression is known to be regulated by *org-1* during specification of alary musculature (Boukhatmi et al., 2014). Here, we find that a parallel mechanism regulates continued specification of niche cells and directs their anterior assembly. We show that *org-1* is expressed in the embryonic niche and reveal its requirement in establishing a full complement of functional, assembled niche cells. We further show that Org1 accumulation is affected by Slit and FGF signals, and that *islet* acts downstream of *org-1*. These data reveal *org-1* as a previously unreported regulator of niche identity and assembly for the *Drosophila* testis niche.

## RESULTS
### *org-1* is expressed in the niche and induced by Slit and FGF
Our previous work showed that the conserved transcription factor *islet* is expressed in the developing niche (Anllo and DiNardo, 2022). Because Org1 is a known *islet* regulator in other tissues (Boukhatmi et al., 2014), and single nuclear RNA sequence data suggest that *org-1* is enriched in the adult testis niche (Li et al., 2022; Raz et al. 2023), we tested whether *org-1* was expressed in the

developing gonadal niche. Using an Org1::GFP transgenic BAC reporter that rescues *org-1* lethality (Kudron et al., 2018; see Materials and Methods), we observed Org1 in prospective niche cells beginning before assembly at embryonic stage 15/16 and after assembly in stage 17 at the gonad anterior (Fig. 1A-C′). Org1::GFP detection in some prospective niche cells coincided with the timing of Tj downregulation (Fig. 1A′,B′). Additionally, we also saw SGPs without downregulated Tj that exhibited lower levels of Org1::GFP elsewhere in the gonad (Fig. 1B-B″, yellow arrows). We confirmed the stage 17 Org1::GFP expression pattern using an Org1 antibody (Schaub et al., 2012) that revealed a signal in gonadal niche cell nuclei (Fig. 1D,D′) that was absent in *org-1* mutants (Fig. S1). Our findings reveal that *org-1* is expressed in the gonad niche, and its expression begins before niche assembly.

Since *islet* is regulated by Slit and FGF from visceral muscle (Vm), we tested if *org-1* is also regulated by these signals (Anllo and DiNardo, 2022). If *org-1* is induced by these signals, then loss of Slit and FGF Vm signals required for niche assembly would result in loss of Org1 accumulation in niche cells. We assessed Org1 accumulation in double mutant animals with the *slit[2]* mutation and the small chromosomal deficiency Df(2R)BSC25 that removes both FGF Heartless ligands: *pyramis* and *thisbe* (see Materials and Methods). Indeed, loss of both signals together resulted in diminished Org1 accumulation (Fig. 1E-F′). Additionally, loss of either Slit or FGF separately resulted in diminished Org1 accumulation (Fig. 1G-K). Together, our data confirm that *org-1* is expressed in the niche and accumulates in response to Slit and FGF.

### *org-1* is necessary and sufficient for aspects of niche cell identity
Given the known role of *org-1* in embryonic muscle specification, we asked if there were niche identity defects in *org-1* mutants (Schaub and Frasch, 2013; Boukhatmi et al., 2014). We hypothesized that if *org-1* had a role in specifying niche fate, we would see disruption of identity markers. We assayed for accumulation and expression, respectively, of the niche identity markers Fasciclin 3 (Fas3) and *hedgehog* (*hh*) (Le Bras and Van Doren, 2006; Wingert and DiNardo, 2015) in an *org-1* allele lacking the translation initiation codon and T-box domain (Schaub et al., 2012). *org-1* mutants did not accumulate the niche-specific adhesion protein Fasciclin 3 (Fig. 2B,B′). Additionally, *org-1* mutants had lower expression of a *hh*-lacZ reporter in niche cells compared to controls (Fig. 2C-E). Interestingly, *hh* expression is known to be restricted to the adult niche (Amoyel et al., 2013). We found that while *hh* expression is highest in control embryonic niche cells, we detected low levels of *hh* in some control SGPs outside the niche (Fig. S2). Since these niche identity markers were disrupted, we used N-cadherin (N-cad) accumulation to detect niche cells in *org-1* mutants. N-cad accumulation is enriched at the cortex of assembled anterior niche cells, with low accumulation in other SGPs, affording a reasonable measure of identification of niche cells at the anterior (Le Bras and Van Doren, 2006). When visualizing N-cad, we detected fewer N-cad accumulating niche cells in *org-1* mutants compared to controls (Fig. 2F-G′). The detection of N-cad in some cells indicates that not all aspects of niche identity are disrupted, suggesting a role for *org-1* in continued induction of niche fate after their initial specification.

To confirm that the fewer N-cad accumulating niche cells in *org-1* mutants resulted from failure of continued niche cell specification in *org-1* mutants, and not simply a reduction in the number of SGPs, we quantified the total number of SGPs in *org-1* mutants. We assayed the SGP-specific transcription factor Tj to count all SGPs (Wingert and DiNardo, 2015). While there was a

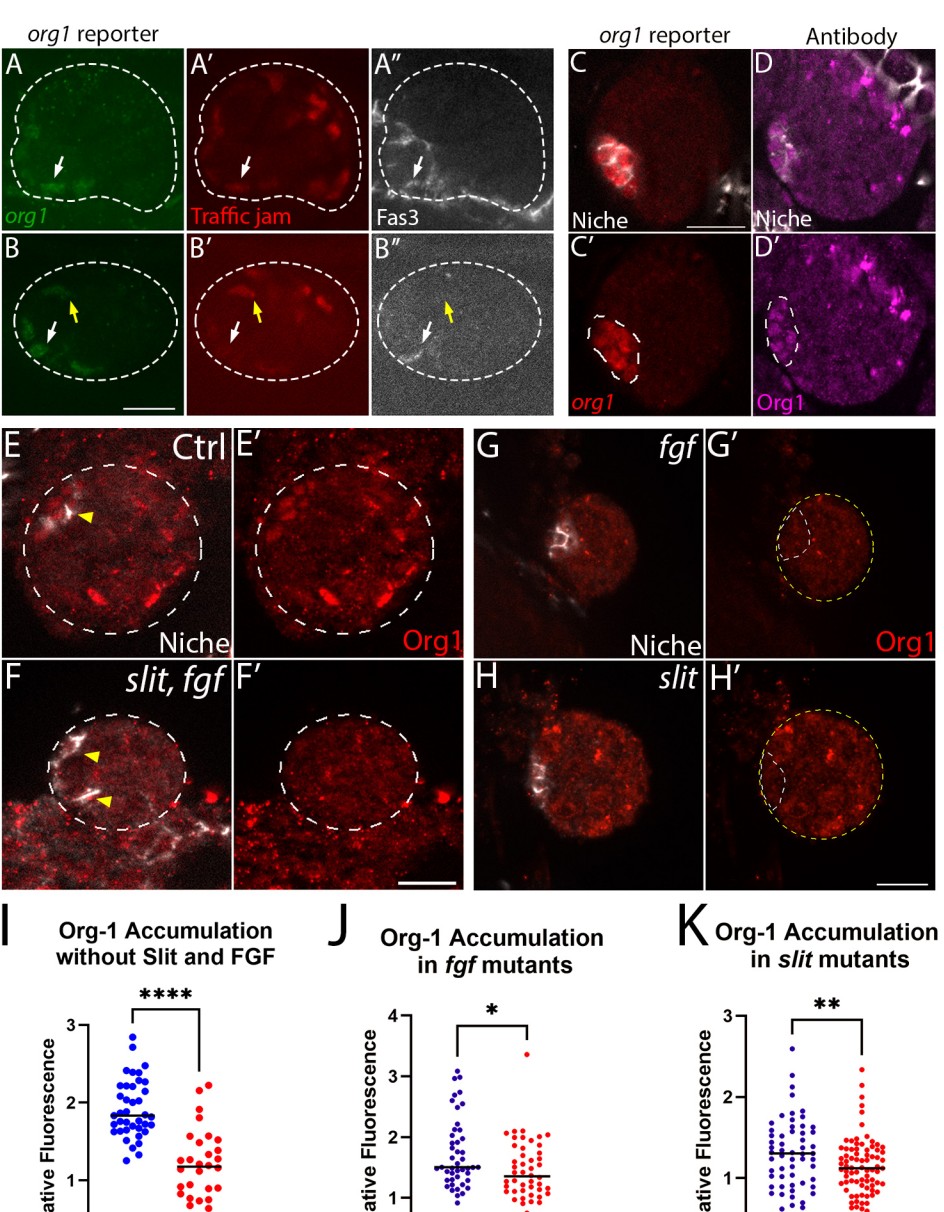

**Fig. 1. *org-1* is expressed in the niche in response to Slit and FGF signals.**
(A-B″) Gonads (dotted lines) from stage 15 animals with the Org1::GFP BAC reporter, immunostained for GFP (green) and Traffic jam (Tj; red). Fas3 (white) accumulates at assembling niche cell interfaces. Org1::GFP is brightest when Tj (red) is downregulated. Prospective niche cells with downregulated Tj (white arrows) and with Tj not downregulated (yellow arrows). (C-H′) Gonads dissected from stage 17 animals aged 20-22 h at 25°C. (C,C′) The Org1::GFP reporter (red) showed *org-1* expression in the assembled niche (dotted white line; Fas3, white). (D,D′) Org1 antibody (magenta) revealed accumulation in the niche (dotted white line; Fas3, white) with spurious non-specific signal at the posterior (see Fig. S1). (E-H′) Dissected stage 17 gonads immunostained to reveal Org1 (red) and the Niche (Fas3, white). (E,E′) Control gonads showed Org1 (red) accumulation in the niche (white and yellow arrows). (F,F′) *slit, fgf* mutants had diminished accumulation of Org1. Gonads are outlined with dotted lines. (G,G′) *fgf* mutants had diminished niche Org1 accumulation. (H,H′) *slit* mutants had diminished niche Org1 accumulation. In G′,H′, gonads are outlined in yellow; niches are outlined in white. (I) Quantification of Org-1 accumulation in niche cells in controls and Slit and FGF signaling mutants (****$P<0.0001$; Ctrl, *n*=39 niche cells from 13 gonads; *slit, fgf*, *n*=30 niche cells from 10 gonads; Mann–Whitney test). (J) Quantified Org1 accumulation in *fgf* mutants and heterozygous sibling controls (Het) (*$P=0.0136$; Het, *n*=45 niche cells from 15 gonads; *fgf*, *n*=48 niche cells from 16 gonads; Mann–Whitney test). (K) Quantified Org1 accumulation in *slit* mutants and heterozygous sibling controls (Het) (**$P=0.0090$; Het, *n*=57 niche cells from 19 gonads; *slit*, *n*=81 niche cells from 27 gonads; Mann–Whitney test). All quantifications are shown for individual niche nuclei. Scale bars: 10 μm. *n*≥3 trials.

slight decrease in number of SGPs, the fraction of these cells specified into niche cells was significantly lower in *org-1* mutants compared to control gonads (Fig. S3). Together with our data indicating a loss of the niche-specific adhesion protein Fas3 and reduction of *hh* gene expression in the niche, these results support a role for *org-1* in continued specification of niche identity (Fig. 2, Fig. S3). Interestingly, *org-1* mutant niche cells still downregulated *tj* as in normal niches, indicating that some aspects of niche specification occur (Fig. S3).

Since *org-1* mutants had fewer niche cells, we tested whether overexpression of *org-1* in all SGPs would create more niche cells. Indeed, when overexpressing *org-1* with a somatic lineage-specific driver (*six4*-gal4; Anllo et al., 2019), we generated additional N-cad- and Islet-accumulating niche cells at the gonad anterior (Fig. 2I-K, see Fig. 5E-F). Together, our data suggest that *org-1* plays a role in specifying niche identity and is sufficient to induce niche identity in additional SGPs.

## *org-1* is required SGP intrinsically for niche identity and assembly

The data above were collected from a whole organism *org-1* mutant line in which all T-box-binding domains were excised (Schaub et al., 2012). To test whether *org-1* is required autonomously in SGPs, we disrupted *org-1* using two somatic lineage-specific drivers that varied in strength: *six4*VP16-gal4 (Warder et al., 2024) and *six4*-gal4. Using the weaker driver (*six4*-gal4), we found an expected decrease in niche cell number, and Fas3 accumulation appeared reduced (Fig. S4). Assuming this driver generates only a partial knockdown, this suggests a stringent requirement of *org-1* in niche cell specification (see Discussion). Excitingly, using the stronger driver (*six4*VP16-gal4), we saw both fewer niche cells (Fig. 3A-C) and dispersed niche cells accumulating low levels of Fas3 (Fig. 3B′,B″,D). It is possible this dispersed phenotype was not something we could observe in the global *org-1* mutants due to the complete loss of Fas3 (Fig. 2). Other niche cell markers, including

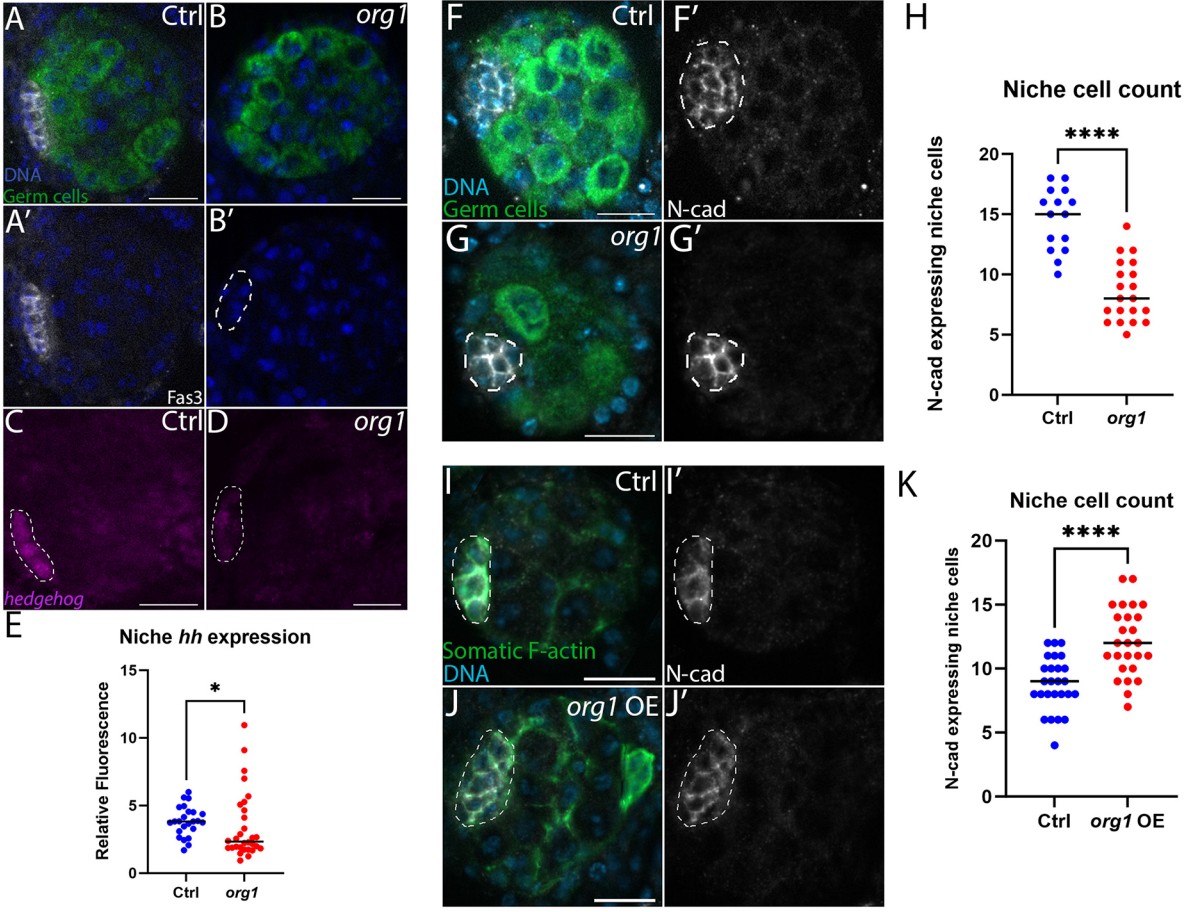

**Fig. 2. *org-1* is necessary and sufficient for aspects of niche cell identity.** (A-D) Dissected gonads immunostained to show germ cells (green), Fas3 (white), *hh*-lacZ (magenta) and DNA (blue). (A,A′,C) Control niche cells accumulated niche specific markers Fas3 and *hedgehog* reporter expression (*hh*-lacZ, separate gonad from A,A′). (B,B′,D) *org-1* mutants did not accumulate Fas3 and had diminished expression of *hh*-lacZ (D is a separate gonad from B). (E) Quantified *hh*-lacZ expression in niche cells accumulating N-cad (**P*=0.0294; Ctrl, *n*=24 niche cells from 7 gonads; *org-1*, *n*=30 niche cells from 10 gonads; Mann–Whitney test). (F,F′) Control gonads accumulated enriched N-cad at the cortex of niche cells. (G,G′) *org-1* mutant niches had fewer cells accumulating enriched N-cad. (F-G′) Germ cells (green), N-cad (white) and DNA (blue). (H) Quantification of N-cad-accumulating niche cells per gonad (*****P*<0.0001; Ctrl, *n*=15 gonads; *org-1*, *n*=20 gonads; Mann–Whitney test). (I,I′) Control gonad with *six4*moesinGFP marking somatic F-actin. (J,J′) SGP-driven *org-1* overexpression in gonads generated niches that had more cells in comparison to controls. (I-J′) Somatic cells (green), niche (white) and DNA (blue). (K) Quantification of niche cell count (*****P*<0.0001; Ctrl, *n*=26 gonads; *org-1*, *n*=26 gonads; Mann–Whitney test). Dotted lines outline the niche. Scale bars: 10 μm. *n*≥3 trials.

E-cad and N-cad, accumulate on control non-niche SGPs to a degree, so dispersed niche cells might not have been apparent (Fig. S5). Our data suggest that *org-1* is primarily a specification factor during niche establishment and secondarily affects niche assembly.

To further support an SGP intrinsic requirement for *org-1* during niche establishment, we performed a tissue-specific rescue experiment. We expressed *org-1* with our somatic lineage-specific driver, *six4*-gal4, in whole organism *org-1* mutants and observed rescued Fas3 accumulation (Fig. 3E-H′). Our rescue experiment simultaneously expressed the F-actin label Ftractin-tdTomato in SGPs, allowing analysis of niche cytoskeletal polarity. Our previous work has shown that F-actin polarizes to niche-stem cell interfaces after assembly is complete (Anllo and DiNardo, 2022; Warder et al., 2024). While F-actin polarization at niche-stem cell interfaces is disrupted in *org-1* mutant niche cells, this polarization was rescued with *org-1* overexpression in SGPs (Fig. 3F-K, see arrows). These data suggest that *org-1* in SGPs is sufficient to initiate downstream cell biological events that polarize F-actin in the assembled niche.

## *org-1* is important for niche to stem cell communication

Given the defects in niche identity and organization seen when *org-1* is compromised, we tested if the function was compromised without *org-1*. We used known niche signaling pathways and centrosome position to assay for niche function deficits (Leatherman and DiNardo, 2010; Kiger et al., 2001; Tulina and Matunis, 2001; Yamashita et al., 2003, 2005). Stat accumulates in GSCs in response to the niche signal Upd1 (Chen et al., 2018; Kiger et al., 2001). We assessed *upd1* expression in niche cells by quantifying levels of *upd1*-Gal4-driven RFP fluorescence and found significantly reduced expression in niche cells in whole organism *org-1* mutants (Fig. 4A-C). Additionally, we measured reduced Stat accumulation in GSCs (Fig. 4D-F), which was consistent with loss of *upd1* expression and impaired niche-GSC communication. Niche cells also send Hh signals to prospective CySCs to enable self-renewal (Michel et al., 2012; Amoyel et al., 2013). We examined localization of Patched, a Hh receptor that accumulates in vesicles in the niche and CySCs in response to signaling (Amoyel et al., 2013; Michel et al., 2012; Wingert and DiNardo, 2015) (Fig. 4G-H′). *org-1* mutants had significantly

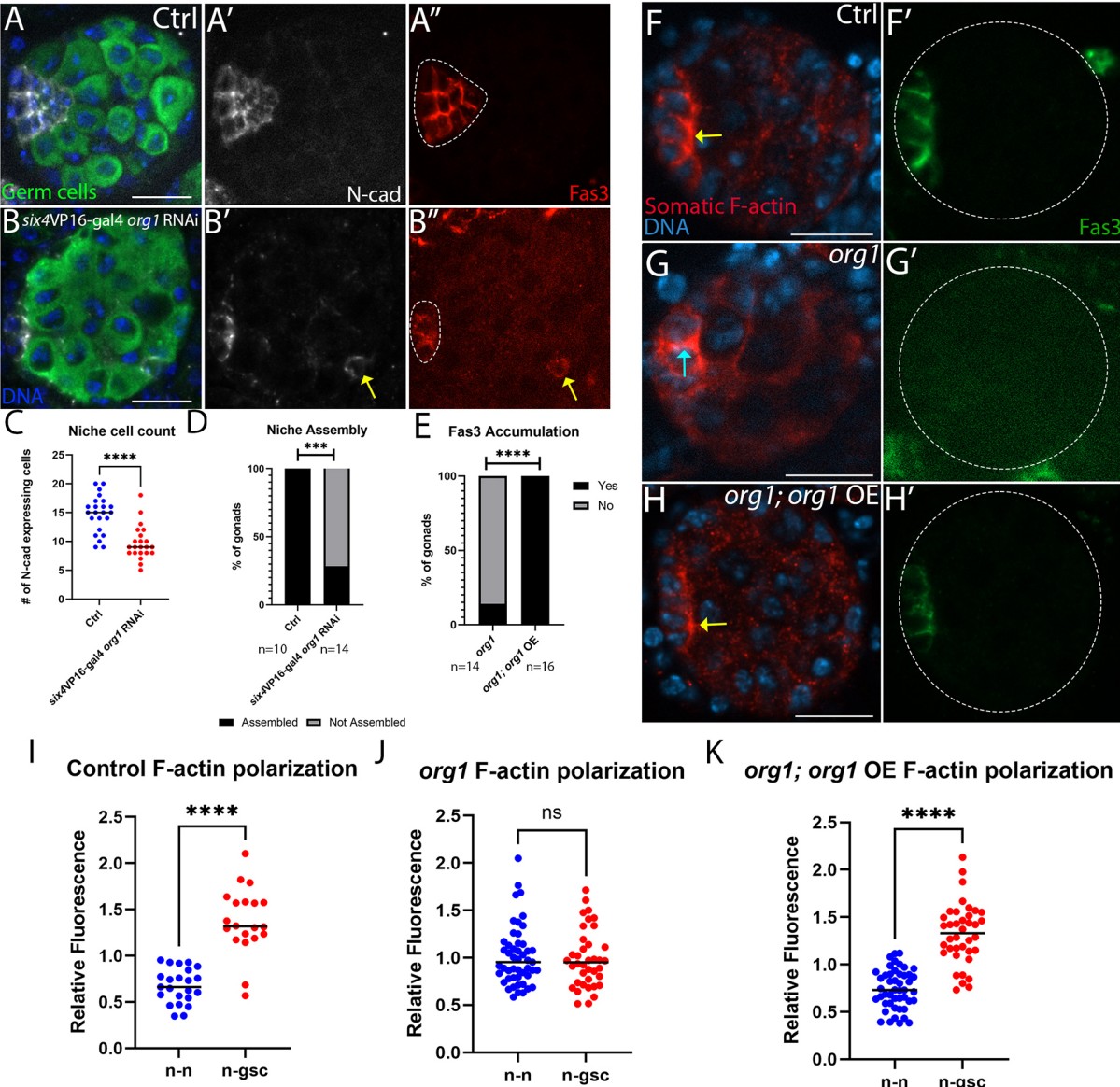

**Fig. 3. *org-1* is required SGP intrinsically for niche identity and assembly.** (A-A″) Control gonads with N-cad (white) and Fas3 (red) accumulating at the anterior. (B-B″) *org-1* knockdown with six4gal4VP16 gonads showed fewer niche cells that accumulate N-cad (white) and Fas3 (red) at the anterior, with displaced niche cells (yellow arrows). Brightness and contrast were adjusted in A″ and B″ to enable visualization of Fas3. (C) Niche cell count per gonad in *org-1* knockdown (****P<0.0001; Ctrl, *n*=23 gonads; six4VP16-gal4 *org-1* RNAi, *n*=22 gonads; Mann–Whitney test). (D) Quantification of niche assembly in six4VP16-gal4 *org-1* RNAi (***P=0.0006; Ctrl, *n*=10 gonads; *org-1* RNAi, *n*=14 gonads; Fisher's Exact test). (E) Quantified Fas3 accumulation in *org-1* rescue niches (****P<0.0001; *org-1*, *n*=14 gonads; *org-1; org-1* OE, *n*=16 gonads; Fisher's Exact test). (F-H′) Yellow arrows indicate enrichment at niche-GSC F-actin interfaces and blue arrows indicate enrichment at niche-niche F-actin interfaces. (F,F′) Control gonad with Fas3 (green) accumulating at the anterior. (G,G′) *org-1* knockout mutant gonad with no Fas3 (green) accumulation. (H,H′) *org-1* knockout mutant with *org-1* overexpression in somatic gonadal cells (red) with rescued Fas3 (green) accumulation. (I) Quantified F-actin polarization in control gonads (****P<0.0001; n-n interfaces, *n*=23 from 7 gonads; n-gsc interfaces, *n*=20 from 7 gonads; Mann–Whitney test). (J) Quantified F-actin polarization in *org-1* mutants (not significant; n-n interfaces, *n*=49 from 14 gonads; n-gsc interfaces, *n*=38 from 14 gonads; Mann–Whitney test). (K) Quantified F-actin polarization in *org-1* mutant rescued gonads (****P<0.0001, n-n interfaces, *n*=47 from 16 gonads; n-gsc interfaces, *n*=38 from 16 gonads; Mann–Whitney test). Scale bars: 10 µm. *n*≥3 trials.

reduced Patched accumulation in the niche and never accumulated Patched in nearby somatic cells (Fig. 4G-I). This finding aligns with the reduced *hh* expression we detected in niche cells when *org-1* was disrupted (Fig. 2C-E), correlating reduced ligand expression to disruption in niche-CySC communication.

We also tested stem cell behavior in the absence of *org-1*. During normal GSC divisions, one centrosome is anchored close to the niche-germline stem cell interface, while the other centrosome moves to the opposite GSC pole. The alignment of one centrosome at the niche interface enables GSC divisions perpendicular to the niche (Yamashita et al., 2003). Gonads from *org-1* null animals showed defects in centrosome alignment, with both GSC centrosomes often displaced from the interface with the nearby niche cell. This displacement positioned centrosomes in orientations that would preclude perpendicular GSC divisions from the niche (Fig. 4J-K′,L). Together, our data suggest *org-1* is required to generate niche cells capable of communication with stem cells, further supporting a role for *org-1* in niche cell specification.

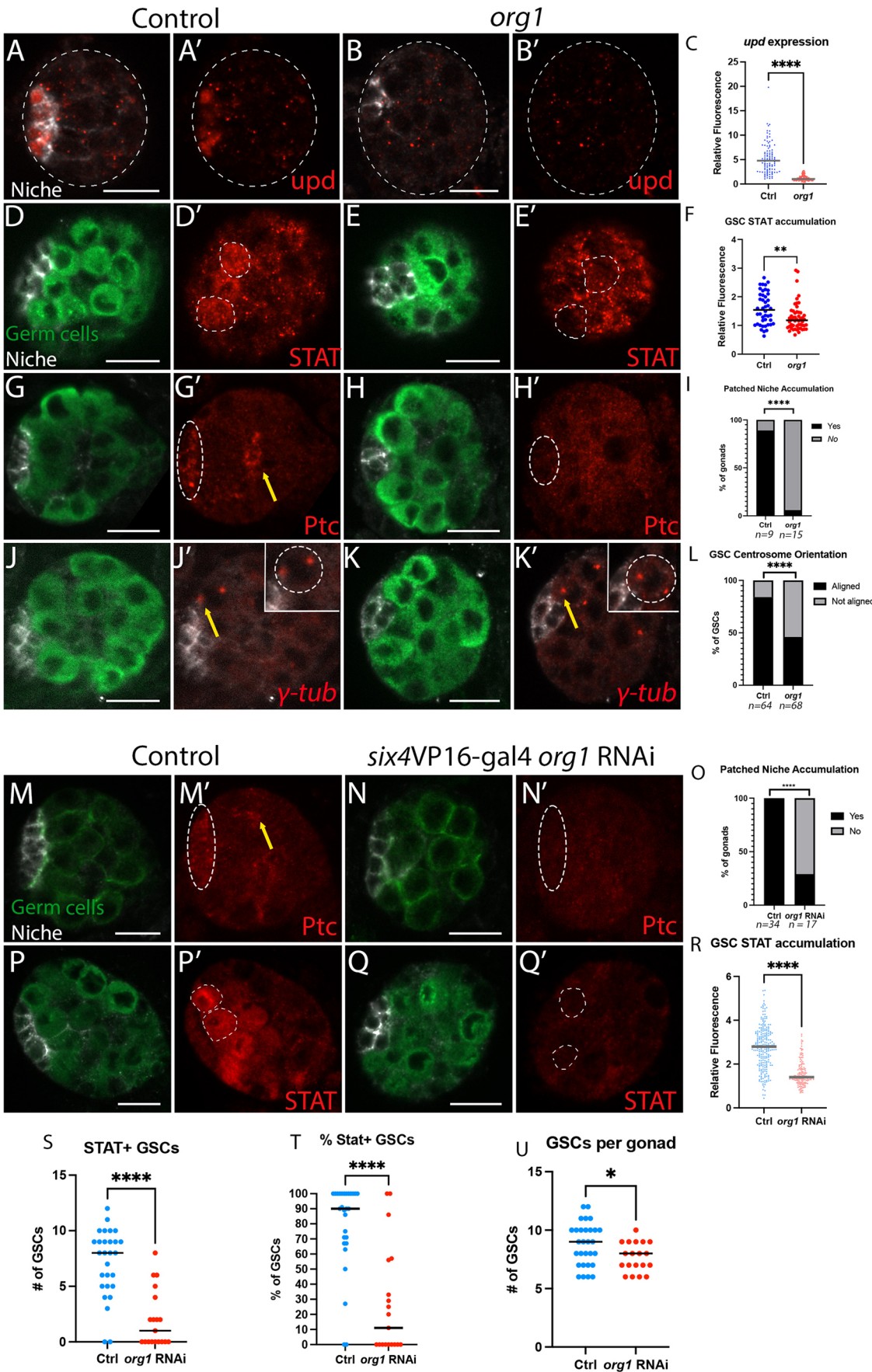

**Fig. 4.** See next page for legend.

**Fig. 4. *org-1* is important for niche to stem cell communication.**
(A-B′) *upd1*-Gal4 drives RFP expression in control niche cells (A,A′), but not in *org-1* null animals (B,B′). (C) Quantification of RFP fluorescence in niche cells (****$P$<0.0001; Ctrl, $n$=100 niche cells from 20 gonads; *org-1*, $n$=140 niche cells from 28 gonads). (D-E′) STAT (red) accumulated in germline stem cells (GSCs, dotted line) in control gonads (D,D′), but not in *org-1* mutant germline stem cells (E,E′). (F) Quantification of STAT accumulation (**$P$=0.0041; Ctrl, $n$=44 GSCs from 11 gonads; *org-1*, $n$=52 GSCs from 16 gonads; Mann–Whitney test). (G,G′) Hedgehog signaling target Patched (red) accumulated in the niche (white dotted line) and cyst stem cells (yellow arrow). (H,H′) Patched did not accumulate in *org-1* mutant gonads in the niche or cyst stem cells. (I) Quantified Patched accumulation in the niche (****$P$<0.0001; Ctrl, $n$=9 gonads; *org-1*, $n$=15 gonads; Mann–Whitney test). (J,J′) In control gonads with duplicated GSC centrosomes, one centrosome was aligned adjacent to the interface with the niche (yellow arrow). Insets show germline stem cells outlined in white, and the niche boundary (Fas3, white). (K,K′) In *org-1* mutant gonads, duplicated centrosomes were often not aligned adjacent to the niche interface (yellow arrow). (L) Quantified centrosome alignment (****$P$<0.0001; Ctrl, $n$=64 GSCs from 18 gonads; *org-1*, $n$=68 GSCs from 21 gonads; Fisher's Exact test). (J′,K′) Insets show centrosome pairs. (M-N′) Hedgehog signaling target Patched accumulated in niche cells and CySCs in controls (M,M′) but not in six4VP16-gal4 *org-1* RNAi gonads (N,N′). Yellow arrow indicates Patched near cyst stem cells. (O) Quantified Patched accumulation in the niche (****$P$<0.0001; Ctrl, $n$=34 gonads; *org-1* RNAi, $n$=17 gonads; Mann–Whitney test). (P-Q′) STAT (red) accumulated in GSCs (dotted line) in control gonads (P,P′) but not in six4VP16-gal4 *org-1* RNAi germline stem cells (Q,Q′). (R-T) Quantified STAT accumulation (R), STAT⁺ GSCs (S) and percentage of GSCs that are STAT⁺ (T) (****$P$<0.0001; Ctrl, $n$=246 GSCs from 28 gonads; *org-1* RNAi, $n$=146 GSCs from 19 gonads; Mann–Whitney test). (U) Number of GSCs contacting N-cad-positive niche cells in control and six4VP16-gal4 *org-1* RNAi gonads (*$P$=0.033; Ctrl, $n$=29 gonads; six4VP16-gal4 *org-1* RNAi, $n$=19 gonads). N-cad is in white. Germ cells are indicated with *nos*-lifeactin::tdTomato (green in M-N′) or Vasa (green, all other panels). $n$≥3 trials. Scale bars: 10 µm.

To investigate whether *org-1* is required cell autonomously in the somatic gonad to enable niche communication with the stem cells, we assessed STAT and Patched accumulation in the six4VP16-gal4 *org-1* RNAi condition. Consistent with *org-1* null animals, somatic gonadal knockdown with *org-1* RNAi resulted in failed accumulation of Patched in the niche and neighboring somatic cells (Fig. 4M-O), and failed accumulation of STAT in neighboring GSCs (Fig. 4P-R). These data indicate a cell-autonomous requirement for *org-1* in communication with stem cells.

We hypothesized that failed communication to neighboring stem cells would adversely affect stem cell maintenance. Neither *org-1* null nor six4VP16-gal4 *org-1* RNAi animals are viable after L1, precluding analysis of later stem cell maintenance. We thus assessed GSC maintenance by quantifying the number of STAT⁺ GSCs adherent to the embryonic niche. We determined a threshold level of STAT⁺ accumulation quantified as one standard deviation below the mean in control GSCs (see Materials and Methods). Our data revealed significantly fewer STAT⁺ GSCs (Fig. 4S,T) and total GSCs adherent to the niche (Fig. 4U) in the six4VP16-gal4 *org-1* RNAi somatic cell knockdown, which disrupts *org-1* during niche specification, and prior to and through niche assembly. These data suggest functional defects in GSC maintenance when *org-1* is autonomously disrupted in the somatic gonad.

### *islet* regulates niche cell specification downstream of *org-1*
Our previous work has shown that *islet* is expressed downstream of Vm signals Slit and FGF, and is important for niche assembly (Anllo and DiNardo, 2022). Here, we show *org-1* is also expressed downstream of Slit and FGF and has a role in niche assembly (see Figs 1 and 3). Because *org-1* is known to induce *islet* expression in embryonic muscle (Boukhatmi et al., 2014), we next tested if *org-1*

regulated *islet* in the embryonic gonad. Indeed, we saw that Islet was no longer enriched in *org-1* mutant niche cells (Fig. 5A-C). We also hypothesized that if *org-1* is regulating *islet*, we would see Islet accumulation in the additional niche cells specified with our SGP lineage-specific *org-1* overexpression. Indeed, we saw that additional niche cells at the anterior also accumulated Islet (Fig. 4D-F). These data indicate that Islet accumulates in response to *org-1* expression.

The accumulation of Islet in response to *org-1* parallels the transcriptional cascade that specifies embryonic alary muscle specification (Boukhatmi et al., 2014). We thus hypothesized that *islet* may also play a role in continued specification of niche identity in addition to the role we previously found during assembly (Anllo and DiNardo, 2022). To test this hypothesis, we quantified the number of Fas3-accumulating niche cells in *islet* mutants compared to controls. Indeed, we counted fewer Fas3-accumulating cells in *islet* mutants, supporting our hypothesis (Fig. 5G-I). This reduction in Fas3-accumulating cells is less striking than the complete loss of Fas3 accumulation in the majority of *org-1* mutant gonads (Fig. 2A-B′, Fig. 3E), suggesting a more prominent role for *org-1* in continued niche specification.

To further support a role for *islet* in niche specification downstream of *org-1*, we investigated whether we could rescue *org-1* mutant niche cell number with *islet* overexpression. We overexpressed *islet* with our SGP lineage-specific driver six4-gal4. In gonads with both the *org-1* mutation and SGP-specific restoration of *islet*, we saw increased numbers of niche cells compared to *org-1* mutants alone (Fig. 5J-M), supporting an ability to rescue niche cell identity. While niche cell number increased with *islet* overexpression, this number was still not fully rescued to control levels, and *islet* overexpression in control genetic backgrounds does not significantly increase niche cell number (Fig. 5M). These results suggest that while *islet* clearly contributes to niche identity, its expression alone is not fully sufficient to induce niche cell fate. The partial rescue also suggests that *islet* is not the only relevant *org-1* target for continued specification. Taken together, our results support a role for *islet* in niche cell specification and suggest that it acts downstream of *org-1* in this role.

### DISCUSSION
An impressive body of work has explored stem cell and niche functions (Anllo and DiNardo, 2022; Kaestner, 2019; Losick et al., 2011; Morrison and Spradling, 2008; Sato et al., 2011; Vida et al., 2025; Warder et al., 2024; Wei et al., 2023; Yamashita et al., 2005), yet how niche cells acquire their identity and are positioned to properly regulate resident stem cells is largely unknown. Prior to our work, Notch signaling and a downstream transcriptional regulator, *tj*, were identified as inputs to *Drosophila* testis niche identity (Kitadate and Kobayashi, 2010; Okegbe and DiNardo, 2011; Wingert and DiNardo, 2015). These inputs are first induced by early contact with the developing gut (Okegbe and DiNardo, 2011). Our previous work studying extrinsic signals important for niche establishment linked niche assembly with function in this system (Anllo and DiNardo, 2022), and prior work has linked niche cell identity and assembly (Wingert and DiNardo, 2015). Here, we have identified a novel regulator that further supports a connection between the processes of niche identity and assembly: the Tbx1 ortholog *org-1*. We show *org-1* is expressed in response to Slit and FGF signals, which we previously identified as niche assembly cues from visceral muscle (Anllo and DiNardo, 2022; this work). In addition, we defined *islet* as a regulator of niche identity operating downstream of *org-1* (Fig. 6, model). The combination of these factors suggests that niche assembly and identity are controlled through overlapping processes. Our work addresses a knowledge gap in mechanisms establishing a functional stem cell niche.

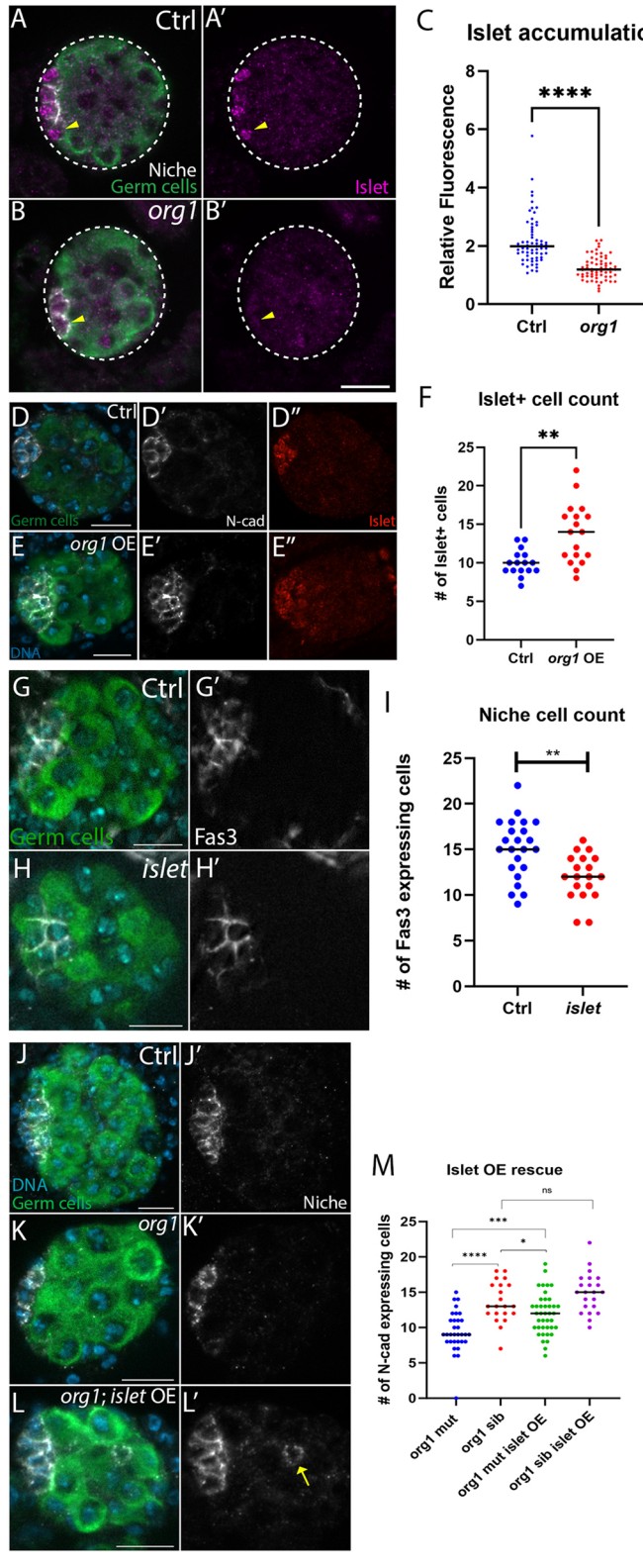

**Fig. 5. *islet* regulates niche cell specification downstream of *org-1*.**
(A,B) Germ cells (Vasa, green), Niche (Ncad, white) and DNA (Hoechst, blue). (A,A′) Control gonads showed Islet (magenta) accumulation enriched in the niche. (B,B′) *org-1* mutant gonads exhibited diminished Islet accumulation. (C) Islet accumulation quantification in niche cells (****P<0.0001; Ctrl, n=63 from 21 gonads; *org-1*, n=63 from 21 gonads; Mann–Whitney test). (D-D″) Control gonads with N-cad (white) and Islet (red) accumulating in niche cells. (E-E″) *org-1* overexpressing gonad with additional niche cells (white) that also accumulated Islet (red). (F) Quantification of the number of Islet accumulating cells (**P=0.0013; Ctrl, n=16 gonads; *org-1* OE, n=18 gonads; Mann–Whitney test). (G,G′) Control gonads with niche (Fas3, white) at the anterior. (H,H′) *islet* mutant gonad with disrupted niche (Fas3, white). (I) Quantification of niche cells per gonad in *islet* mutants (**P=0.0019; Ctrl, n=23 gonads; *islet*, n=19 gonads; Mann–Whitney test). (J,J′) Control gonads with N-cad (white) and Islet (red) accumulating in niche cells. (K,K′) *org-1* mutant gonad with fewer niche cells accumulating N-cad at the anterior. (L,L′) An *org-1* mutant with *islet* overexpression in somatic gonadal cells had more niche cells than *org-1* mutants, and a displaced niche cell (yellow arrow). (M) Number of niche cells per gonad for *org-1* rescue with *islet* overexpression experiments: *org-1* mut versus *org-1* sib, ****P<0.0001 (*org-1* mut, n=32 gonads; *org-1* sib, n=21 gonads); *org-1* mut versus *org-1* mut *islet* OE, ***P=0.0005 (*org-1* mut, n=32 gonads; *org-1* mut *islet* OE, n=40 gonads); *org-1* sib versus *org-1* mut *islet* OE, *P=0.0338 (*org-1* sib, n=21 gonads; *org-1* mut *islet* OE, n=40 gonads); and *org-1* sib versus *org-1* sib *islet* OE, not significant (ns) (*org-1* sib, n=21 gonads; *org-1* sib *islet* OE, n=22 gonads) (Mann–Whitney tests). Scale bars: 10 μm. n≥3 trials.

between niche identity and assembly (Wingert and DiNardo, 2015). In response to the niche specification signal Notch, the transcription factor *tj* is downregulated. *tj* mutants specify additional niche cells that do not express all niche identity markers, and as a result do not assemble at the anterior (Wingert and DiNardo, 2015). Interestingly, activating other requirements of niche identity, such as Bowl, places the additional *tj* mutant niche cells at the gonad anterior (Wingert and DiNardo, 2015; DiNardo et al., 2011). This suggests that reduction of *tj* alone is not sufficient for niche assembly at the anterior, supporting a role for additional regulators specifying niche cells capable of assembly. Perhaps the signals necessary for anterior assembly (Slit and FGF) are unable to be received by ectopic niche cells in *tj* mutants.

We have shown that niche assembly signals Slit and FGF are important for regulating the novel niche identity regulator *org-1*. Gonads with compromised *org-1* show disruptions to markers of niche identity, including complete loss of Fas3, reduction of *hh*, loss of *upd1* expression, and fewer cells accumulating N-cadherin. The few niche cells that accumulate N-cad exhibit assembly defects in which some of these cells do not arrive at the anterior (this work). Loss of continued specification after earlier Notch signaling might account for incomplete assembly when *org-1* is disrupted. *tj* mutants show additional cells expressing some niche identity markers, and some of these cells are located in positions away from the gonad anterior (Wingert and DiNardo, 2015). Together with work presented here, these data link proper specification of niche identity with anterior assembly. While these data support that niche identity is important for aspects of niche assembly, some niche cells still assemble anteriorly in instances when continued specification is disrupted. Perhaps differential adhesive sorting enables partially specified pro-niche cells at the gonad anterior to assemble.

Interestingly, loss of *slit* and *fgf* regulators does not completely mimic phenotypes seen in *org-1* mutants. For example, *slit* and *fgf* mutants accumulate Fas3, whereas *org-1* mutants do not. We detect residual Org1 accumulation in niche cells from *slit* and *fgf* mutants (Fig. 1), suggesting that additional regulators are impacting *org-1* expression. We suspect that while Slit and FGF both regulate *org-1*,

## Testis niche identity and assembly are linked

Our previous work showed that the visceral muscle signals Slit and FGF are required for niche assembly. When niche assembly was compromised, niche function was disrupted, even though these manipulations did not directly affect niche identity (Anllo and DiNardo, 2022). Previous work has suggested a connection

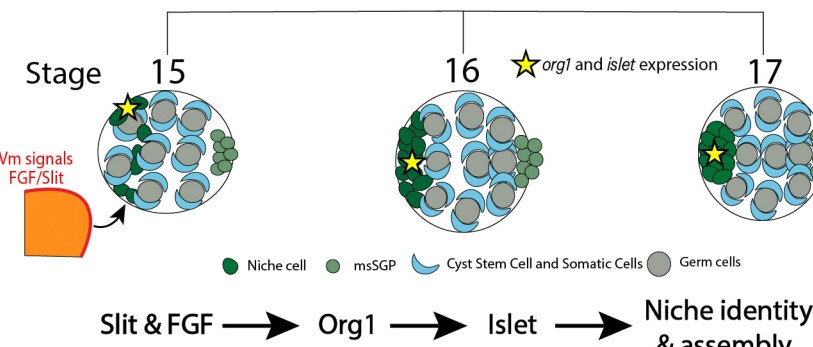

**Fig. 6. Predicted model of establishment of niche identity and assembly.** Diffusible signals Slit and FGF regulate *org-1*. The visceral muscle (Vm) is a neighboring tissue source of both signals. *org-1* functions to regulate *islet* for niche identity and assembly.

Slit and FGF signaling have a primary role facilitating niche cell anterior placement, while *org-1* has a primary function in continued specification of niche identity.

### *islet* has dual roles in niche identity and assembly downstream of *org-1*

To further link niche identity and assembly, we present an exciting new role for the transcription factor *islet* in niche specification in addition to the role we previously revealed for *islet* in niche assembly. Together with our previous study, this work reveals that Slit and FGF signals induce both *org-1* and *islet* expression in niche cells (Anllo and DiNardo, 2022; this work). The visceral muscle (Vm) is an exciting candidate source of the Slit and FGF ligands inducing *org-1*, as previous work revealed Vm as an important tissue supplying these signals for niche assembly (Anllo and DiNardo, 2022). In addition, we show that *org-1* regulates *islet* in the niche, as *org-1* mutant niches lack Islet, and niche cell numbers are increased by overexpressing *islet* in SGPs in an *org-1* mutant background. These data support the model of a transcriptional cascade regulating *islet* first through Slit and FGF signals inducing *org-1*, which in turn induces *islet* expression (Fig. 6). Based on this model, we would expect *islet* loss in the *org-1* null background would not further disrupt niche specification, but genetic odds of conducting this experiment were prohibitive. *org-1* has a more prominent role in continued niche specification than *islet*, as *org-1* mutants lack Fas3 accumulation (Fig. 2A-B′; Fig. 3E), whereas *islet* mutants have fewer Fas3 positive niche cells (Fig. 5I). This result, along with the finding that *islet* overexpression only partially rescues niche cell number (Fig. 5M), suggests other genes act downstream of *org-1* in addition to *islet*. Interesting remaining tasks include identifying these additional *org-1* targets that contribute to continued specification, and revealing how *islet* influences the number of niche cells that accumulate Fas3 and N-cad.

The detection of dispersed, unassembled niche cells in our *six4*VP16-gal4 driven *org-1* RNAi was unexpected. It is reasonable to suggest earlier and stronger knockdown driving *org-1* RNAi with the enhanced *six4*VP16-Gal4 driver compared to that observed with *six4*-Gal4. Since we observed only assembly defects under the presumed stronger conditions, we infer that *org-1* has a primary role in niche cell identity and a secondary role in niche assembly.

### *org-1* regulates aspects of testis niche identity

In alignment with the conserved role of Tbx1 orthologs in specifying mesoderm, we found it fitting to test if *org-1* had a role in specifying the mesodermally derived niche cells. Using the niche identity markers Fasciclin3 (Fas3), *unpaired1* (*upd1*) and *hedgehog* (*hh*), we saw that *org-1* mutants were unable to properly accumulate these markers (Le Bras and Van Doren, 2006; Okegbe and DiNardo, 2011). *hh* is linked to niche assembly downstream of

Notch (Wingert and DiNardo, 2015). Using the *hh*-lacZ reporter, we detected high *hh* expression in the niche, and low expression in SGPs outside the niche (Fig. S2). While we find that *hh* expression is not solely restricted to the niche in the embryonic gonad as it is in the adult (Amoyel et al., 2013), our ability to detect higher *hh* expression in niche cells enables us to use its upregulated expression as a reliable marker for niche identity.

There are residual niche cells in *org-1* mutants, which suggests that other inputs of niche identity are partly active. It has been previously described that Notch signaling is necessary for specifying niche cells (Kitadate and Kobayashi, 2010; Okegbe and DiNardo, 2011). The average number of niche cells in Notch mutants is far fewer than we see in *org-1* mutants, which suggested to us that *org-1* is either one of multiple effectors downstream of Notch or is involved in a parallel niche specification mechanism. One of the effects of Notch signaling in specifying niche cells is to downregulate Tj accumulation in these cells. Previously, it was not known when downregulation of *tj* began in prospective niche cells (Wingert and DiNardo, 2015). We detected downregulation of *tj* coinciding with the beginning of *org-1* expression in prospective niche cells after gonad coalescence. Since *org-1*-expressing cells had lowered levels of Tj, we considered whether *org-1* might repress Tj. However, we have shown that Tj is not regulated by *org-1* (Fig. S2). Alternatively, Tj might block *org-1* expression. This relationship could explain the higher accumulation or Org1::GFP we detect in SGPs with reduced Tj (Fig. 1B-B″, yellow arrows). Future work should assess whether Tj regulates *org-1* downstream of Notch.

Our work has established *org-1* as a previously unreported regulator of identity and assembly during testis niche establishment. Factors identified in the *Drosophila* testis stem cell niche have translated into other systems, including a role for Notch in niche specification (Zamfirescu et al., 2022). Because of the translatable cell biological concepts elucidated in this niche, our work lays the foundation for future studies to investigate a conserved capacity of Tbx1 to specify gonadal mesoderm or niche cell fate.

### Study limitations

While we reveal a role for *org-1* in continued niche specification during development, flies with our genetic manipulations do not survive past embryogenesis. This precludes our ability to assess later effects of losing *org-1* during development, and it is possible that aspects such as niche formation are simply delayed. It is also formally possible that defects in niche formation are secondary effects caused by lethality. We do not believe the latter to be the case because we do not observe gross morphological defects or problems with visceral muscle morphogenesis in late-stage *org-1* RNAi embryos. This work importantly reveals a role for *org-1* during initial niche formation, and future work will be required to study a

role for *org-1* in adult niche maintenance. Finally, while assessing double mutant animals would be one way to support evidence of *islet* downstream of *org-1*, genetic odds of conducting this experiment are not feasible.

## MATERIALS AND METHODS
### Embryonic gonad dissection and immunostaining
Embryos were collected for 2 h and then aged for 20-22 h in a humidified chamber at 25°C to stage 17. Embryos were then sorted for mutant and control genotypes using fluorescently marked balancer chromosomes (FM7DfdYFP or TM6DfdGFP). Sorted embryos were dissected and immunostained as previously described (Anllo et al., 2019). Dissected tissue was incubated at 4°C overnight with primary antibodies as follows: goat anti-Vasa [1:800; Santa Cruz Biotech (discontinued)], rabbit anti-Vasa (1:5000; BosterBio, DZ41154), mouse anti-Fasciclin III (1:50; DSHB AB_528238), rat anti-N-cadherin (1:20; DSHB AB_528121), rat anti-E-cad (1:20; DSHB AB_528120), rabbit anti-Stat92E (1:200; Flaherty et al., 2010), rabbit anti-RFP (1:500; Abcam ab62341), mouse anti-Islet (1:200; DSHB AB_528313), guinea pig anti-Traffic jam (1:10,000; a gift from D. Godt, University of Toronto, Canada), rat anti Org-1 (1:200; a gift from C. Schaub, EMBL, Heidelberg, Germany), mouse anti-Gamma Tubulin (1:200; Sigma AB_477584), chicken anti-Beta-gal (1:1000; Abcam ab9361-250), mouse anti-Patched (1:50; DSHB AB_528441) and chicken anti-GFP (1:1000; Aves labs AB_2307313). Secondary antibodies were used at 3.75 µg/ml (Alexa488, Cy3 or Alexa647; Molecular Probes; Jackson Immunoresearch) for 1.5-2 h at room temperature. DNA was stained with Hoechst 33342 (Sigma) at 0.2-0.3 µg/ml for 6 min.

Tissue was imaging using a Zeiss AxioImager.Z1 microscope equipped with and ApoTome.2, X-cite Xylis fluorescent light source and Axiocam 705 mono camera, using a 40×1.2 N.A. plan apo lens. Org1::GFP reporter images in stage 15 embryos were obtained with a Crest X-Light V3 spinning disk confocal on an Olympus IX83 platform with a 60×1.3 N.A. plan apo silicone oil immersion objective.

### Visualization of stage 15 embryonic gonads
Embryos were collected on grape juice agar plates supplemented with a dab of fresh yeast paste. After allowing adult flies to lay fertilized eggs for 16 h, the embryos were transferred to a nylon mesh strainer by rinsing with de-ionized water and using a paintbrush. Embryos were de-chorionated with 50% bleach for no longer than 5 min. Embryos were fixed in a glass vial at 23°C in a 1:1 mixture of 4% paraformaldehyde (PFA) in PBS:heptane for 18 min while on a nutator (Mitchison and Sedat, 1983). The PFA was removed and replaced with an equal volume of 100% methanol. The vial was vigorously shaken for 30-60 s to devitellinize the embryos. Heptane was then removed and devitellinized embryos were washed with 100% methanol three times. Embryos were washed once with 50% methanol in phosphate-buffered saline (PBS), then again with 100% PBS with 0.1% Triton for gentle rehydration. Embryos were blocked for at least 1 h in 4% normal donkey serum, prior to immunostaining and imaging as described above. Stage 15 embryos were selected for imaging based on Campos-Ortega and Hartenstein (1985).

### *Drosophila* strains
Genotypes for all tissue are listed in Table S1. All *Drosophila* lines used are listed in Table S2. *org-1[OJ487]* is a null allele lacking the translation initiation codon and T-box domain (Schaub et al., 2012). Org1::GFP is a bacterial artificial chromosome (BAC) including the *org-1*-coding sequence and nearby genetic regulatory elements upstream of an in-frame GFP sequence (Kudron et al., 2018). We have confirmed that this BAC rescues *org-1* lethality by generating a stable fly line in which all viable adults are homozygous for *org-1[OJ487]* and have the org1::GFP BAC transgene. *slit[2]* is a null allele with no detectable protein product (Battye et al., 2001; Nusslein-Volhard et al., 1984). To remove *pyr* and *ths* FGF ligands together, we used a small chromosomal deficiency, Df(2R)BSC25, that completely deletes the genes encoding both ligands that bind the Heartless FGF receptor (Stathopoulos et al., 2004). When generating *org-1[OJ487]*, *upd1*-Gal4 recombinant chromosomes, we screened for homozygous lethality and the ability to drive *upd1*-Gal4, UAS-redStinger expression in larval tissues.

### Identification of male gonads and the niche
We identified male gonads by the presence of male-specific SGPs at the posterior (DeFalco et al., 2003), which were visualized with Vasa (BosterBio) or six4moeGFP (Sano et al., 2012). Additionally, male gonads can be identified by the fact that they have more proliferative germ cells and therefore have more germ cells than female gonads (Anllo et al., 2019; Wawersik et al., 2005). Niches in stage 17 embryos were identified via immunofluorescence by niche cell-specific accumulation of Fasciclin 3 (DSHB AB_528238), or enrichment of N-cadherin (DSHB AB_528121) or E-cadherin (DSHB AB_528120).

### Characterization of niche phenotypes
When assessing morphology and proper assembly of the niche, we characterized a niche as either 'assembled' or 'unassembled'. 'Assembled' niches show all niche cells contacting one another at the gonad anterior and have clearly smoothened cell boundaries around the niche periphery. 'Unassembled' niches have niche cells anterior and at the gonad periphery, but with irregular boundaries, and/or have dispersed niche cells that are not contacting the anterior cluster of niche cells.

### Rigor and reproducibility
All experiments were conducted at least in triplicate ($n \geq 3$ trials) with sufficiently large sample sizes that previously enabled detection of significant differences (significance level of less than or equal to 0.05%). Genetic controls were always used, comparing perturbed conditions to the most genetically similar control background feasible.

### Quantification of niche cell count
When quantifying the number of cells that express N-cad, Fas3 or any other niche marker, we use the ImageJ Cell Counter Plugin to record cells that we counted manually. A niche cell will accumulate Fas3 when in contact with other niche cells and will accumulate N-cad on all interfaces. Niche cells that are counted with either Fas3 or N-cad are co-labeled with a nuclear marker such as Tj or Hoechst, to enable clear identification of distinct cells. Dispersed niche cells that accumulate Fas3 do not show polarized enrichment to specific cell interfaces.

### Quantification of normalized niche cell expression/accumulation
When quantifying accumulation of proteins and reporters of gene expression, we used ImageJ software to measure mean gray value fluorescence intensity within regions of interest (ROIs). We selected ROIs including a circular region within somatic cell boundaries, using either Fas3 or N-cad immunofluorescence to delineate niche cell boundaries and Hoechst dye to delineate nuclei. ROIs were in a single *z* plane in which the relevant cell was in focus. For all experiments except *upd* measurements, three niche cell ROIs were measured from each gonad. For *upd* measurements, five niche cell ROIs were measured per gonad. To quantify background fluorescence, a ROI was selected to encompass the unlabeled region of a single germ cell within each gonad, and a separate background ROI drawn from a region where no tissue was present. Background-subtracted fluorescence values for each niche cell were normalized, dividing by the background-subtracted value from the neighboring unlabeled germ cell for the respective gonad. Relative fluorescence values for each niche cell were plotted. Mann–Whitney tests were used to evaluate comparisons.

### Quantification of fluorescence at cell junctions
To quantify fluorescence intensity of F-actin between cell interfaces, we used ImageJ software to measure mean gray value fluorescence intensity within regions of interested (ROIs). ROIs were either niche-niche interfaces or niche-GSC interfaces. We selected 4-pixel-wide segmented line ROIs between niche cells and between niche cells and GSCs. Two to four niche-niche and niche-GSC interfaces were measured in every gonad using N-cad to visualize each interface. We measured the mean fluorescence intensity of our Ftractin-tdTomato blindly with N-cad as a boundary. To estimate background fluorescence, a separate ROI was drawn from a region where no tissue was present. Background-subtracted fluorescence values for each

niche cell were normalized, divided by the background-subtracted average value of all cell interfaces. Relative fluorescence values for each interface were plotted. Mann–Whitney tests were used to evaluate comparisons.

## Quantification of stat accumulation

To quantify Stat accumulation, we stained gonads using an anti-Stat antibody (1:200; E. Bach, NYU, USA) and used ImageJ to measure the mean gray value fluorescence intensity within ROIs. During dissections and staining, we used the phosphatase inhibitor PhosStop (Sigma, 4906845001). We selected ROIs including a circular region to sample germ cells, using Vasa immunofluorescence as a marker to delineate cell boundaries. Whole germ cells were measured in a single $z$ plane in which the relevant cell was in focus. We selected ROIs by only using the germ cell marker to mark sample regions to avoid biased observations towards brighter or dimmer fluorescence of Stat. For each gonad, we sampled all the GSCs we could clearly identify and 3-7 neighboring germ cells in the second tier, further from the germline stem cells. After background subtraction, we measured the ratio of Stat accumulation within each GSC relative to the neighboring germ cell average for that gonad. Relative Stat enrichment values were plotted for each GSC. We obtained measurements on mutants for *org-1* and their sibling controls, and *org-1* RNAi knockdowns with their sibling controls. Mann–Whitney tests were used to evaluate comparisons.

## Quantification of STAT⁺ GSCs

To determine whether a germ cell adherent to the niche was STAT[+], we first counted the number of germline stem cells (GSCs) directly adjacent to N-cadherin-labeled niche cells. We then quantified the mean fluorescence intensity from each of these adherent germ cells as described above, measuring mean STAT enrichment for each adherent germ cell in *org-1* RNAi knockdowns and sibling control animals. We established a threshold STAT[+] fluorescent value, measured as one standard deviation below the relative mean fluorescence intensity of controls. For controls, the relative mean fluorescence intensity was 2.76, with a standard deviation of 0.97; therefore, any adherent germ cells with a mean fluorescence intensity above 1.79 were counted as STAT[+] in both control and *org-1* RNAi knockdown gonads. Data were imported into Prism v10.0 to generate scatterplots where each dot represents a gonad. We plotted the number of STAT[+] GSCs per gonad, and separately the percentage of STAT[+] GSCs per gonad for *org-1* RNAi and sibling control conditions. A Mann–Whitney test was used to evaluate the comparison.

## Centrosome anchoring quantification

Centrosome position was visualized with immunofluorescence against gamma tubulin to label pericentriolar material. GSCs were scored for centrosome position if they had already undergone centrosome duplication, indicating that they had advanced at least to the G2 stage of the cell cycle (Chen et al., 2018). We quantified how often one of the two centrosomes was positioned closer to the adjacent niche, marked by N-cad, than to other neighboring cells. GSCs with a centrosome located near the niche-GSC interface were scored as appropriately aligned. GSCs with both centrosomes positioned away from the niche were scored as misaligned. Significance was assessed with Fisher's Exact test.

### Acknowledgements
We acknowledge Kirklan Naumuk for technical assistance. We thank Drs Elizabeth Ables and Beth Thompson, and the Lenhart lab for valuable feedback on the manuscript. We thank the Bloomington *Drosophila* Stock Center (NIH P40OD018537) for stocks, and D. Godt, E. Bach and C. Schaub for antibodies. We thank the ECU Imaging Core Facility.

### Competing interests
The authors declare no competing or financial interests.

### Author contributions
Conceptualization: P.H., L.A.; Data curation: P.H., L.A.; Formal analysis: P.H., A.H., T.G., L.A.; Funding acquisition: S.D., L.A.; Investigation: P.H., T.G., L.A.; Methodology: L.A.; Project administration: S.D., L.A.; Resources: S.D., L.A.; Supervision: S.D., L.A.; Validation: P.H., L.A.; Visualization: P.H., A.H., L.A.; Writing – original draft: P.H.; Writing – review & editing: P.H., A.H., S.D., L.A.

### Funding
This work was supported by the National Institutes of Health (R15 GM154246 to L.A. and R35 GM136270 to S.D.). Open Access funding provided by East Carolina University. Deposited in PMC for immediate release.

### Data and resource availability
All relevant data and details of resources can be found within the article and its supplementary information.

### Peer review history
The peer review history is available online at https://journals.biologists.com/dev/lookup/doi/10.1242/dev.204914.reviewer-comments.pdf

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
