## [Peer Review File · Development (Cambridge, England)]

The Tbx1 ortholog *org-1* is required to establish testis stem cell niche identity in *Drosophila*

Patrick Hofe, Ariel Harrington, Tynan Gardner, Stephen DiNardo and Lauren Anllo
DOI: 10.1242/dev.204914

Editor: Swathi Arur

Review timeline

Original submission:	5 May 2025
Editorial decision:	4 June 2025
First revision received:	16 October 2025
Editorial decision:	10 November 2025
Second revision received:	20 November 2025
Accepted:	2 December 2025

Original submission

First decision letter

MS ID#: dev.204914

MS TITLE: Tbx1 ortholog *org-1* is required to establish testis stem cell niche identity in *Drosophila*

AUTHORS: Patrick Hofe, Ariel Harrington, Tynan Gardner, Stephen DiNardo and Lauren Anllo

Dear Dr Anllo,

I have now received all the referees' reports on the above manuscript, and have reached a decision. The referees' comments are appended below, or you can access them online: please go to:

As you will see, the referees express considerable interest in your work, but provide recommendations which will enhance the rigor and analysis of your manuscript. If you are able to revise the manuscript along the lines suggested, which may involve further experiments, I will be happy receive a revised version of the manuscript. Your revised paper will be re-reviewed by one or more of the original referees, and acceptance of your manuscript will depend on your addressing satisfactorily the reviewers' major concerns. Please also note that Development will normally permit only one round of major revision.

Please attend to all of the reviewers' comments and ensure that you clearly highlight all changes made in the revised manuscript. Please avoid using 'Tracked changes' in Word files as these are lost in PDF conversion. I should be grateful if you would also provide a point-by-point response detailing how you have dealt with the points raised by the reviewers in the 'Response to Reviewers' box. If you do not agree with any of their criticisms or suggestions please explain clearly why this is so.

Reviewer 1

Advance summary and potential significance to field

Manuscript dev.204914 by Hofe et al explores the role of the Tbx1 ortholog, *org1*, in regulation of *Drosophila* testis stem cell niche formation. Study of the molecular mechanisms controlling testis

stem cell niches in *Drosophila* has led to greater understanding of stem cell niche developing in general; revealing key paradigms controlling this process that are conserved in other organisms. This work is, therefore, highly relevant to your journal's readership.

The author's identification and analysis of *org1*'s role in testis stem cell niche formation is also important. Given prior findings that *islet* is required for testis niche assembly, and that *islet* is regulated by *org1* in developing musculature, the observation that *org1* plays a role in testis stem cell niche formation is, perhaps, not surprising. Still, confirmation was necessary and this work represents a careful analysis of *org1*'s role in promoting niche cell identity and assembly during testes development. Furthermore, dissecting the role of *org1* in different aspects of niche formation (including specification of niche cell fate, induction of niche cell identity, and the assembly of niche cells into a functional hub) will provide important insights into the steps and mechanisms controlling these distinct processes in other organisms.

Comments for the author

1) Suggest reviewing use of the term "niche specification" and revising manuscript for clarity. At times, the authors refer to niche specification as the induction of niche cell fate from anterior SGPs that is regulated by Delta/Notch signaling. At other times, niche specification appears to refer to the continued acquisition of niche cell identity that occurs during establishment of a functional niche at the end of embryogenesis. While specification is a continuum leading up to formation of a functional stem cell niche, broad use of the term "specification" for different stages of niche formation that occur at different times in development can be confusing. A few examples in the text that cause confusion are listed below. To alleviate this concern, it is suggested that the author's settle on specific terminology that distinguishes different stages of niche cell fate acquisition. For example, the authors might use "specification" to refer to the initial acquisition of niche cell identity from SGPs, while they might use "establishment" "induction" to refer to the continued acquisition of niche cell identity. Alternatively, they might use "initial specification" and "continued specification" to distinguish between the initial acquisition of niche cell fate and further induction of niche cell identity, respectively. However this is done, the authors should be careful to be consistent as to how they do this throughout the paper. This language could be carefully defined in the introduction and then used throughout the paper.

Some specific lines of text causing confusion due to ambiguous language::

-Line 158: "To confirm that fewer N-Cadherin cells in *org1* mutants resulted from failure to specific niche cells and not simply a reduction..." (does this refer to initial specification or continued acquisition of identity???)

-Line 164: "... these results support a role for *org1* in niche specification (again, initial specification or continued acquisition of identity?)

-Lines 63-67: Once specified, embryonic niche cell are compartmentalized to the gonad anterior in a process called niche assembly. During assembly, specified pro-niche SGPs move...." (this isn't bad in and of itself because it refers to pro-niche cells, but add on Figure 7 where Niche assembly and Niche identity are indicated to both occur at stage 17, and it becomes confusing)

-Line 242: "... may also play a role in specifying niche cells in addition to the role we previously identified during assembly" (again, initial specification working, or later specification steps?)

-Lines 274: "... a knowledge gap in mechanisms specifying function stem cell niches, revealing signals that control niche identity and proper assembly..." (perhaps better here would be establishing a functional stem cell niche)

2) Did the authors examine the effects of *org1* loss of function or somatic RNAi knockdown in later stages of development? *org1* inhibition shows a clear effect on niche identity and assembly in late stage embryos, but it would be interesting to know if this is caused by an inability of these processes to occur correctly, or if the effects are simply caused by a delay in these processes so that a functional niche is established in larval testes rather than at the end of embryogenesis.

3) The authors nicely show that *org1* acts autonomously in the somatic gonad to promote induction of niche identity as well as assembly. It would be nice if they could extend at least some of these experiments to determine if it is also required cell autonomously for niche communication with the stem cells. While it might be considered beyond the scope of this paper, it would also be

interesting if functional analyses were performed to assess effects on germline stem cell and cyst stem cell maintenance by counting the somatic and germline stem cell numbers in *org1* mutants.

4) Suggest being more cautious of about the statement "*Org1* is necessary and sufficient to induce niche cell fate" (lines 137 & 538). Given that DN-Cadherin is expressed in the newly formed niche, and the authors even use it to count hub cell number, this indicates that "aspects" of niche cell identity are established. The authors do state that "some aspects of niche specification occur", (again, that pesky specification term when talking about late stage embryos)... but the authors could make this clearer in their section/figure titles. For example: "*org1* is necessary and sufficient for aspects of niche cell identity" or "*org1* is necessary and sufficient for normal niche cell fate induction."

5) The authors should be clearer about which stages of development they are discussing. This is especially the case in Figure 1 where they show images of stage 15 embryos/animals, as well as the "assembled niche". The methods section does state that 20-22 hour embryos were analyzed, so it is implied that images in panels C-H are from these late stage embryos, but it should be stated explicitly in both the figure legend and text.

6) Methods should include an explanation of how stage 15 embryos were obtained. This is currently not described in the methods.

Additional minor comments:

1) Walker et al, 2008 reference (lines 29 & 30) is not found in references (though Walker et al, 2009 is present).

2) Where are the scale bars in Figure 1 E-H? I think they may be in the bottom right corner of F' and H', with the scale bar in F' matching all panels of E-F and the scale bar in H' matching with all panels in G-H' but the figure organization was confusing... especially since the testes in panels G,G' look much smaller (is this the mutation, or is this scaling?)

Reviewer 2

Advance summary and potential significance to field

An understanding of how testis stem cell niche identity is specified has been a surprisingly difficult "nut to crack" and so the work presented here is significant. True, *org1* does not appear to be required for all aspects of niche identity, as hub cell number is decreased, rather than eliminated, in *org1* null mutants, and a hub still seems to form, but it seems to clearly play a role. Further, placing *org1* upstream of *islet* in this process is also an important finding. The work presented is fairly compact—short and sweet—which probably fits a shorter "report" format better than the "article" format. For example, Figs 2+3 make very related points and could be combined into one plus supplemental, similarly for Figs 5+6, and a model fig is not necessary. In addition, there are some major and minor concerns that need to be addressed. However, since these should all be able to be addressed, I think the work could be appropriate for a Development Report.

Major concerns

I'm a bit confused about the phenotype. Some hub markers, like *fas3* and *hh* are down, but *N-cad* is still present and the hub still seems to form. They need to describe this better in the text and perhaps experimentally. For example, if hub markers are off, but the cells still undergo hub morphogenesis, what does this say about the process of hub formation? Can they look at the hub later (perhaps in their RNAi condition) to see if it is functional?

In figure 2, the gain of function experiment needs to be done more completely as it is important. The use of *N-cad* is confusing, since it was not greatly decreased in hub cells that still form in *org1* mutants. At least one additional hub marker needs to be used.

In figure 4, a simple explanation for why hub-GSC communication is affected is if ligand expression is reduced. They demonstrate this for *hh*, but never link this to the *ptc*-reporter result. *upd* expression should also be examined. Further, if ligand expression is down because hub identity is down, then these results seem a bit obvious. Lastly, my understanding of the "centrosome assay" as

used in the past is not the distance of the "hub-proximal" centrosome to the niche but whether the orientation of the centrosomes is perpendicular to the niche or random. They should reanalyze these data accordingly.

What is the nature of the islet mutant phenotype in relation to *org1*? It appears weaker, which would indicate that *islet* is not the only gene downstream of *org1*. They should analyze and describe both phenotypes with this in mind. In addition, another test of their "linear" model between *org1* and *islet* is that the double-mutant *islet/org1* should resemble, and be no stronger than, *org1* alone. Is this the case?

Minor concerns

--Overall, the figures are not completely labeled or described in text, leading reader to have to search for answers. This is very frustrating and makes the reader concerned about the quality of the work. Examples include: what is the *Org1* reporter?, What is the niche marker in Fig 1E-G? What is an "FGF" mutant if there are 3 FGFs in flies? What is the control genotype in Fig 3A?

-the *Org1* antibody panel is mis-identified in the text

-panels 2A and 2A' do not appear identical

-the authors indicate that the *slit/fgf* experiment indicates that signals from the visceral mesoderm are required for *org1* expression. While this may be true, they cannot claim this from a whole-animal double mutant, as these factors are present in other places. Therefore, they should change their conclusions to reflect this.

-what is the control genotype for the *six4-VP16* RNAi and OE experiments? It needs to be the *six4VP16* by itself.

First revision

Author response to reviewers' comments

Comments from the Reviewers:

Reviewer 1: SUMMARY OF THE ADVANCE MADE IN THIS PAPER AND ITS POTENTIAL SIGNIFICANCE TO THE FIELD

Manuscript dev.204914 by Hofe et al explores the role of the *Tbx1* ortholog, *org1*, in regulation of *Drosophila* testis stem cell niche formation. Study of the molecular mechanisms controlling testis stem cell niches in *Drosophila* has led to greater understanding of stem cell niche developing in general; revealing key paradigms controlling this process that are conserved in other organisms. This work is, therefore, highly relevant to your journal's readership.

The author's identification and analysis of *org1*'s role in testis stem cell niche formation is also important. Given prior findings that *islet* is required for testis niche assembly, and that *islet* is regulated by *org1* in developing musculature, the observation that *org1* plays a role in testis stem cell niche formation is, perhaps, not surprising. Still, confirmation was necessary and this work represents a careful analysis of *org1*'s role in promoting niche cell identity and assembly during testes development. Furthermore, dissecting the role of *org1* in different aspects of niche formation (including specification of niche cell fate, induction of niche cell identity, and the assembly of niche cells into a functional hub) will provide important insights into the steps and mechanisms controlling these distinct processes in other organisms.

SUGGESTIONS TO AUTHORS

1) Suggest reviewing use of the term "niche specification" and revising manuscript for clarity. At times, the authors refer to niche specification as the induction of niche cell fate from anterior SGP that is regulated by *Delta/Notch* signaling. At other times, niche specification appears to refer to the continued acquisition of niche cell identity that occurs during establishment of a

functional niche at the end of embryogenesis. While specification is a continuum leading up to formation of a functional stem cell niche, broad use of the term "specification" for different stages of niche formation that occur at different times in development can be confusing. A few examples in the text that cause confusion are listed below. To alleviate this concern, it is suggested that the author's settle on specific terminology that distinguishes different stages of niche cell fate acquisition. For example, the authors might use "specification" to refer to the initial acquisition of niche cell identity from SGPs, while they might use "establishment" "induction" to refer to the continued acquisition of niche cell identity. Alternatively, they might use "initial specification" and "continued specification" to distinguish between the initial acquisition of niche cell fate and further induction of niche cell identity, respectively. However this is done, the authors should be careful to be consistent as to how they do this throughout the paper. This language could be carefully defined in the introduction and then used throughout the paper.

This is a helpful suggestion for clarity. We took the reviewers advice and defined "initial specification" and "continued specification" in the introduction (see lines 52, 55-61). We have chosen to use "continued specification" to describe the events we define in this manuscript because at this time of development, Notch signaling has already been active. We have corrected each of the 5 specific lines of text mentioned below to reflect this clarification.

Some specific lines of text causing confusion due to ambiguous language::

-Line 158: "To confirm that fewer N-Cadherin cells in *org1* mutants resulted from failure to specific niche cells and not simply a reduction..." (does this refer to initial specification or continued acquisition of identity???)

-Line 164: "... these results support a role for *org1* in niche specification (again, initial specification or continued acquisition of identity?)

-Lines 63-67: Once specified, embryonic niche cell are compartmentalized to the gonad anterior in a process called niche assembly.

During assembly, specified pro-niche SGPs move...." (this isn't bad in and of itself because it refers to pro-niche cells, but add on

Figure 7 where Niche assembly and Niche identity are indicated to both occur at stage 17, and it becomes confusing)

-Line 242: "... may also play a role in specifying niche cells in addition to the role we previously identified during assembly" (again, initial specification working, or later specification steps?)

-Lines 274: "... a knowledge gap in mechanisms specifying function stem cell niches, revealing signals that control niche identity and proper assembly..." (perhaps better here would be establishing a functional stem cell niche)

2) Did the authors examine the effects of *org1* loss of function or somatic RNAi knockdown in later stages of development? *org1* inhibition shows a clear effect on niche identity and assembly in late stage embryos, but it would be interesting to know if this is caused by an inability of these processes to occur correctly, or if the effects are simply caused by a delay in these processes so that a functional niche is established in larval testes rather than at the end of embryogenesis.

This is an excellent question. We have tried to answer it by looking at both *org1* nulls or *six4VP16-Gal4* driven RNAi against *org1*, but the animals do not survive past larval stage L1. The L3 larval stage is typically evaluated when examining larval testes. This technical limitation impedes our ability to examine older animals that have had *org1* disrupted throughout embryogenesis. There is currently no functional data for a role of the *org1-islet* connection in the adult testis, thus making examination of disruption during adulthood outside the scope of this paper. Our work here uniquely contributes a role for *org1* in the initial construction of this niche.

3) The authors nicely show that *org1* acts autonomously in the somatic gonad to promote induction of niche identity as well as assembly. It would be nice if they could extend at least some of these experiments to determine if it is also required cell autonomously for niche communication with the stem cells. While it might be considered beyond the scope of this paper, it would also be interesting if functional analyses were performed to assess effects on germline stem cell and cyst stem cell maintenance by counting the somatic and germline stem cell numbers in *org1* mutants.

This is an insightful suggestion. For embryonic gonads, there is currently no cyst stem cell (CySC) specific marker to allow counting of these cells. However, in this resubmission we show an effect on germline stem cells upon intrinsic knockdown of *org1* in somatic cells. We assessed GSC maintenance by counting the number of GSCs (defined as GCs contacting the hub), and quantifying the number and percentage of hub-adherent GSCs with above-threshold levels of STAT accumulation in the *org1* RNAi somatic-specific manipulation. This data has been added to both Figure 4 and the same Results section, "*org1* is important for niche to stem cell communication."

Additionally, to extend our analysis to determine cell autonomous requirements for niche communication with stem cells, we have repeated our analysis of STAT and Ptc accumulation with the somatic *org1* RNAi manipulation. This data revealed reduced ability to signal to stem cells, and has been added to Figure 4 and the associated Results section, "*org1* is important for niche to stem cell communication."

4) Suggest being more cautious of about the statement "Org1 is necessary and sufficient to induce niche cell fate" (lines 137 & 538). Given that DN-Cadherin is expressed in the newly formed niche, and the authors even use it to count hub cell number, this indicates that "aspects" of niche cell identity are established. The authors do state that "some aspects of niche specification occur", (again, that pesky specification term when talking about late stage embryos)... but the authors could make this clearer in their section/figure titles. For example: "*org1* is necessary and sufficient for aspects of niche cell identity" or "*org1* is necessary and sufficient for normal niche cell fate induction."

We changed the wording indicated in these two lines (now 161 and 541) to reflect the authors suggestion, "*org1* is necessary and sufficient for aspects of niche cell identity."

5) The authors should be clearer about which stages of development they are discussing. This is especially the case in Figure 1 where they show images of stage 15 embryos/animals, as well as the "assembled niche". The methods section does state that 20-22 hour embryos were analyzed, so it is implied that images in panels C-H are from these late stage embryos, but it should be stated explicitly in both the figure legend and text.

We added clarification to the Figure 1 legend stating that panels C-H represent stage 17 embryos obtained from a 20-22 hour collection at 25 degrees C.

6) Methods should include an explanation of how stage 15 embryos were obtained. This is currently not described in the methods.

Thank you for pointing this out. We added a section to the Methods defining how we process stage 15 embryos for immunostaining, to visualize stage 15 gonads (lines 797-813).

Additional minor comments:

1) Walker et al, 2008 reference (lines 29 & 30) is not found in references (though Walker et al, 2009 is present). The references section was corrected to reflect Walker et al., 2008.

2) Where are the scale bars in Figure 1 E-H? I think they may be in the bottom right corner of F' and H', with the scale bar in F' matching all panels of E-F and the scale bar in H' matching with all panels in G-H' but the figure organization was confusing... especially since the testes in panels G,G' look much smaller (is this the mutation, or is this scaling?). For panels E-H, we added scale bars to each of the prime panels so that each gonad depicted includes a scale bar.

Reviewer 2: An understanding of how testis stem cell niche identity is specified has been a surprisingly difficult "nut to crack" and so the work presented here is significant. True, *org1* does not appear to be required for all aspects of niche identity, as hub cell number is decreased, rather than eliminated, in *org1* null mutants, and a hub still seems to form, but it seems to clearly play a role. Further, placing *org1* upstream of islet in this process is also an important finding. The work presented is fairly compact—short and sweet—which probably fits a shorter "report" format better than the "article" format. For example, Figs 2+3 make very related points and could be combined

into one plus supplemental, similarly for Figs 5+6, and a model fig is not necessary. In addition, there are some major and minor concerns that need to be addressed. However, since these should all be able to be addressed, I think the work could be appropriate for a Development Report.

Major concerns

I'm a bit confused about the phenotype. Some hub markers, like *fas3* and *hh* are down, but *N-cad* is still present and the hub still seems to form. They need to describe this better in the text and perhaps experimentally. For example, if hub markers are off, but the cells still undergo hub morphogenesis, what does this say about the process of hub formation? Can they look at the hub later (perhaps in their RNAi condition) to see if it is functional?

The added description of initial vs continued specification, suggested by Reviewer 1 above, clarifies the presence of *N-cad* despite disruption to other markers, such as *Fas3* and *hh*. We added language clarifying this point in the introduction, and throughout the Results section where suggested by Reviewer 1. We further added language to the Discussion to address the question raised by Reviewer 2 here about the process of hub formation when aspects of niche specification are disrupted (Paragraph 2 of "Testis niche identity and assembly are linked").

The idea to examine the hub later in development in the RNAi condition was also suggested by Reviewer 1 above. We would love to ask this question, but as stated earlier, the animals do not survive to larval stage L3 of development, precluding our ability to do so. Our work importantly reveals a role for *org1* in the initial construction of this niche, and further exploration of a role for *org1* in niche maintenance after development is beyond the scope of this work.

In figure 2, the gain of function experiment needs to be done more completely as it is important. The use of *N-cad* is confusing, since it was not greatly decreased in hub cells that still form in *org1* mutants. At least one additional hub marker needs to be used.

We had included other markers of hub identity in the *org1* overexpression experiments later in the manuscript, after Figure 2. We edited the text to refer to this data (Figure 5) now in the Results section for Figure 2I-K. Note that the gain of function experiment described in Figure 3 already included two other hub markers (accumulation of *Fas3* and polarization of F-actin).

In figure 4, a simple explanation for why hub-GSC communication is affected is if ligand expression is reduced. They demonstrate this for *hh*, but never link this to the *ptc*-reporter result. *upd* expression should also be examined. Further, if ligand expression is down because hub identity is down, then these results seem a bit obvious. Lastly, my understanding of the "centrosome assay" as used in the past is not the distance of the "hub-proximal" centrosome to the niche but whether the orientation of the centrosomes is perpendicular to the niche or random. They should reanalyze these data accordingly.

We added text linking the reduction of *hh* to the *Ptc*-reporter result in the Results section where niche communication to *CySC* is addressed (lines 284-286).

As suggested, we now assessed *upd* expression in *org1* mutants, quantifying this by analyzing *upd-GAL4 > RFP* expression. This new data is added to Figure 4, and the corresponding results section, "org1 is important for niche to stem cell communication."

We have re-analyzed our centrosome data in the suggested more traditional manner, and updated results and Methods sections accordingly.

What is the nature of the *islet* mutant phenotype in relation to *org1*? It appears weaker, which would indicate that *islet* is not the only gene downstream of *org1*. They should analyze and describe both phenotypes with this in mind. In addition, another test of their "linear" model between *org1* and *islet* is that the double-mutant *islet/org1* should resemble, and be no stronger than, *org1* alone. Is this the case?

We agree with the reviewer that the *islet* phenotype appears weaker, suggesting that *islet* is not the only gene downstream of *org1*. We added text to the Results section comparing the *Fas3* phenotype in *islet* mutants to that in *org1* mutants, and directly stating this suggestion when

defining the *islet* overexpression partial rescue of niche cell number (lines 349-351; 359-361). We further addressed this point with additions to the Discussion section referencing *islet* (lines 439-454).

Assessing a double mutant of *islet* and *org1* is not experimentally feasible, with less than 1 in 16 embryos being double mutants and male. Using RNAi for both *Org1* and *Islet* is not a rigorous test since RNAi knockdowns can be partial and thus not informative for epistasis experiments, and, in any case, in our hands we have not found an *islet* RNAi that can work at this stage.

Minor concerns

--Overall, the figures are not completely labeled or described in text, leading reader to have to search for answers. This is very frustrating and makes the reader concerned about the quality of the work. Examples include: what is the *Org1* reporter?, What is the niche marker in Fig 1E-G? What is an "FGF" mutant if there are 3 FGFs in flies? What is the control genotype in Fig 3A?

We described the nature of the *Org1* BAC reporter is in the Methods section defining *Drosophila* strains. For clarity, we added information that this is a BAC reporter to the Figure 1 legend, and the Results text.

We added text to the Figure 1 legend clarifying the Niche marker in Fig1E-G.

The nature of the FGF disruption is described in the *Drosophila* strains of the Methods as the small chromosomal deficiency *Df(2R)BSC25* that completely deletes the genes encoding both *pyr* and *ths*. We added information to the methods clarifying that these are the two FGF ligands that bind the Heartless FGF receptor (line 468). The RRID for the relevant line from the Bloomington *Drosophila* stock center is listed in our Key Resources table. For clarity, we added information about this deficiency to the relevant Results section, lines 136-138.

To clarify control genotypes for each experiment, we have added a Supplemental Table that includes the genotypes for all experiments. The experiment in question in Figure 3A was a genetic cross of female *six4VP16-Gal4 / CyO DfdGFP* with male *org1RNAi / CyO DfdGFP*. Mutant animals (genotype in 3B) were *six4VP16-Gal4 / UAS-org1RNAi*. Control animals were age-matched sibling animals from the same collection that retained the *CyO DfdGFP* balancer, so either driver alone (*six4VP16-Gal4 / CyO DfdGFP*) or RNAi alone (*UAS-org1RNAi / CyO DfdGFP*). We have included this genotype with others in our Supplemental Table.

-the *Org1* antibody panel is mis-identified in the text

We updated the text to reflect the correct *Org1* antibody Figure panel (1D,D').

-panels 2A and 2A' do not appear identical

Thank you for noticing this. These were in fact the same image, just rotated to different angles during image processing to generate figure panels. We have corrected the rotation in the updated version of Figure 2.

-the authors indicate that the slit/fgf experiment indicates that signals from the visceral mesoderm are required for *org1* expression. While this may be true, they cannot claim this from a whole-animal double mutant, as these factors are present in other places. Therefore, they should change their conclusions to reflect this.

We adjusted conclusions stated in Results, Figure titles, and Discussion to reflect the suggested change.

-what is the control genotype for the *six4-VP16* RNAi and OE experiments? It needs to be the *six4VP16* by itself.

Our new Supplemental Table of genotypes now clarifies the genotypes for all experimental and control conditions presented. The control for the *org1* overexpression experiment is the *UAS-org1* alone, with no driver. This accounts for any potential background activity of the *UAS-org1* that is

not driven in a tissue specific manner. The control genotype for the *six4*-VP16 RNAi experiment is either driver alone (*six4*VP16-Gal4 / CyO DfdGFP) or RNAi alone (UAS-*org1* RNAi / CyO DfdGFP). Both the driver and UAS-*org1* RNAi are balanced in trans to the CyO, DfdGFP balancer chromosome, which is the only fluorescent balancer that reliably allows us to genotype embryos without affect gene expression in the gonad. These lines are not viable as homozygous adults.

Second decision letter

MS ID#: dev.204914R1

MS TITLE: Tbx1 ortholog *org-1* is required to establish testis stem cell niche identity in *Drosophila*

AUTHORS: Patrick Hofe, Ariel Harrington, Tynan Gardner, Stephen DiNardo and Lauren Anllo

Dear Dr Anllo,

I have now received all the referees reports on the above manuscript, and have reached a decision. The referees' comments are appended below.

The overall evaluation is positive and we would like to publish a revised manuscript in *Development*, although I will recommend that it be published as a short report, provided that the referees' comments can be satisfactorily addressed. Please attend to all of the reviewers' comments in your revised manuscript and detail them in your point-by-point response. If you do not agree with any of their criticisms or suggestions explain clearly why this is so. If it would be helpful, you are welcome to contact us to discuss your revision in greater detail. Please send us a point-by-point response indicating your plans for addressing the referees' comments, and we will look over this and provide further guidance.

Reviewer 1

Advance summary and potential significance to field

While a succinct manuscript, this paper provides an analysis *org1*'s role in aspects of testis niche formation that should be highly relevant to the readership of *Development*.

Comments for the author

The authors thoughtfully address the majority of comments by both prior reviewers leading to a manuscript that is much approved. A number of minor text recommendations are included below. The authors also address technical limitations that impact their ability to address comments by looking at double mutants or later stages of niche development due to lethality in a "study limitation" section at the end of the discussion. At the same time, failure of the samples to survive past embryogenesis leads to some additional questions related to lethality. Specifically, when do the samples begin to show defects linked to lethality, what are these defects, and is it possible that the defects in late stages of embryonic niche development are secondary effects caused by lethality? This concern would be reduced if the authors do not observe gross morphological problems and/or problems with mesodermal patterning in late stage embryos, or a subset of the samples survive into early stages of larval development so they can be studied. At the very least, however, concerns related to lethality should be addressed in the text section of the study limitations. For example, text might include "This precludes our ability to assess later effects of losing *org1* during development, and it is possible that aspects of niche formation are simply delayed. It is also formally possible that defects in niche formation are due to secondary effects caused by lethality. We do not believe the latter to be the case because...."

So the reader doesn't wait until the last paragraph of the paper to be alerted to limitations, the author might also consider noting some of them in the results and/or main discussion text as well.

Additional, minor text recommendations follow:

Line 11 - consider revising to: "... mutants caused defects in niche assembly and functionality." ... as current text could be interpreted as defects resulted in niche assembly, but caused defects in niche functionality.

Line 40 - Description of the hub as "the singular cell type that comprises the niche" is somewhat debatable as the niche for the GSCs might be considered to be comprised of hub cells as well as the cyst stem cells.

Line 82 - consider synonyms for "maintain" as it is used twice in short succession

Line 373 - Title of this section is a bit overstated. As in results section, the authors should consider softening to "org1 regulates aspects of testis niche identity"

Reviewer 2

Advance summary and potential significance to field

The authors have not done much in the way of experiments to address reviewer comments. They do respond to all the comments, but are mostly "talking their way out of them". Still, as I indicated before, I feel this work is of a quality and level of interest for Development. However, since it is limited in scope, it is more appropriate for a shorter Report rather than an Article.

Second revision

Author response to reviewers' comments

Comments from the Reviewers:

Reviewer 1: SUMMARY OF THE ADVANCE MADE IN THIS PAPER AND ITS POTENTIAL SIGNIFICANCE TO THE FIELD

While a succinct manuscript, this paper provides an analysis of org1's role in aspects of testis niche formation that should be highly relevant to the readership of Development.

SUGGESTIONS TO AUTHORS

The authors thoughtfully address the majority of comments by both prior reviewers leading to a manuscript that is much appreciated. A number of minor text recommendations are included below. The authors also address technical limitations that impact their ability to address comments by looking at double mutants or later stages of niche development due to lethality in a "study limitation" section at the end of the discussion. At the same time, failure of the samples to survive past embryogenesis leads to some additional questions related to lethality. Specifically, when do the samples begin to show defects linked to lethality, what are these defects, and is it possible that the defects in late stages of embryonic niche development are secondary effects caused by lethality? This concern would be reduced if the authors do not observe gross morphological problems and/or problems with mesodermal patterning in late stage embryos, or a subset of the samples survive into early stages of larval development so they can be studied. At the very least, however, concerns related to lethality should be addressed in the text section of the study limitations. For example, text might include "This precludes our ability to assess later effects of losing org1 during development, and it is possible that aspects of niche formation are simply delayed. It is also formally possible that defects in niche formation are due to secondary effects caused by lethality. We do not believe the latter to be the case because...."

This is a thoughtful note. We can reduce the reviewer's concerns by telling them that we do not observe gross morphological defects in our tissue-specific org1 RNAi manipulations. Most of the

experimental animals survive into L1 larval stages, which begins 30-60 min after we dissect late Stage 17 gonads. Lethality occurs between L1 and the later L3 larval stage that the field studies when referencing “larval” gonads.

Of note, we have studied other embryonic lethal mutants with gross mesodermal defects that do not affect gonadogenesis. For example, *jelly belly (jeb)* mutants lack visceral muscle founder cells, and completely lose visceral muscle precursors by Stage 15 of embryogenesis, and these animals form a morphologically normal testis niche (Anllo & DiNardo, 2022, *Dev Cell*). Thus, lethal mutations that affect other embryonic mesoderm do not always lead to effects in the gonadal mesoderm.

We have added the statement suggested by this reviewer to the “Study Limitations” section with this rationale. See lines 495-499.

So the reader doesn't wait until the last paragraph of the paper to be alerted to limitations, the author might also consider noting some of them in the results and/or main discussion text as well.

We have added a reference to study limitations related to prohibitive genetic odds of removing both *org1* and *islet* in the Discussion text in lines 419-420. To compensate for added text while keeping within the 7000 word limit, we removed 97 words from other places throughout the text as noted in the highlighted copy of the revised manuscript.

Additional, minor text recommendations follow:

Line 11 - consider revising to: "... mutants caused defects in niche assembly and functionality." ... as current text could be interpreted as defects resulted in niche assembly, but caused defects in niche functionality.

We have done this.

Line 40 - Description of the hub as "the singular cell type that comprises the niche" is somewhat debatable as the niche for the GSCs might be considered to be comprised of hub cells as well as the cyst stem cells. This is a good point. We have edited the text to not exclude the CySCs as a component of the niche.

Line 82 - consider synonyms for "maintain" as it is used twice in short succession. Thank you for pointing this out. We have varied the word choice here, which is now line 100.

Line 373 - Title of this section is a bit overstated. As in results section, the authors should consider softening to "org1 regulates aspects of testis niche identity". We have made this change, now line 449.

Reviewer 2: The authors have not done much in the way of experiments to address reviewer comments. They do respond to all the comments, but are mostly "talking their way out of them". Still, as I indicated before, I feel this work is of a quality and level of interest for Development. However, since it is limited in scope, it is more appropriate for a shorter Report rather than an Article.

We are sorry that Reviewer #2 does not feel we have done much in the way of experiments. We would like to point out that in revisions, we repeated all niche functionality assays in our tissue specific knockdown, which allowed us to support a tissue specific role for *org1* in signaling to stem cells. We also generated a tricky recombinant X chromosome that allowed us to report on the niche's ability to produce stem cell signals. These experiments added rigor to our work, as they allowed us to directly link defective niche signaling to the disrupted stem cell response. In revisions, we also performed data analysis that allowed us to define a somatic specific requirement for *org1* in maintaining germline stem cells.

We agree with this reviewer that some of our figures can be combined to make the number of figures more manageable. We have combined the former Figures 5 & 6 into a single figure. We were asked to add additional text to clarify our phenotype in our initial round of revisions. This addition brought us over 7000 words, which our second Reviewer #1 noted was still "succinct." To preserve

the clarity that the original reviewers asked us to add, we have kept the text at 7000 words. We feel that this maintains the level of quality of the manuscript as it allows us to more thoughtfully describe our data. Removing 60% of the text to alter to a Short Report would require rewriting the text in a way that risks loss of the clarity we were asked to add in our initial revisions.

Third decision letter

MS ID#: dev.204914R2

MS TITLE: Tbx1 ortholog org-1 is required to establish testis stem cell niche identity in Drosophila

AUTHORS: Patrick Hofe, Ariel Harrington, Tynan Gardner, Stephen DiNardo and Lauren Anllo

Dear Dr Anllo,

I am happy to tell you that your manuscript has been accepted for publication in Development, pending our standard publication integrity checks.